# Optimizing Black Soldier Fly frass fertilizer rates for maize and soybean production in Madagascar

**Tanjona Ramiadantsoa**[1]*, **Anthonely Amady**[2], **Marc Philippart**[3], **Marcellio Tombozara**[2], **Vincent Lucas**[2], **Brian L. Fisher**[1,4]

**1** Madagascar Biodiversity Center, Antananarivo, Madagascar, **2** EXA Feed, Antsiranana, Madagascar, **3** Independent Consultant, Moramanga, Madagascar, **4** California Academy of Sciences, San Francisco, California, United States of America

* tanjona.ramiadantsoa@gmail.com

## Abstract

In low-yield regions, yield intensification rather than land extensification offers the most efficient pathway to increase production and meet rising food demand. Yet, limited access to fertilizers remains a major constraint on crop productivity. The rapid expansion of insect farming has created a growing supply of an insect-derived organic fertilizer produced during insect rearing (frass); however, guidance on optimal field-scale application rates remains scarce. We conducted a large-scale field experiment in northern Madagascar to quantify yield responses of maize and soybean to Black Soldier Fly (*Hermetia illucens*) frass fertilizer and to identify optimal biophysical and economic application rates. Frass was applied at rates ranging from 0 to 30 t ha$^{-1}$. For both crops, we did not identify a fertilizer rate that would maximize yield; instead, dry grain yield always increased with fertilizer rate and approached asymptotic maxima corresponding to 12.0- and 1.7-fold increases relative to unfertilized controls for maize and soybean, respectively. These rates are predicted to produce yields equivalent to 4.6- and 6.7-fold increases over national average yields, respectively. An economic analysis based on the marginal value–cost ratio (MVCR) indicated that the frass price is constrained by the profitability of soybean. Nevertheless, if using the national recommended dose of 3.6 t ha$^{-1}$, a retail price of USD 13.8 t$^{-1}$ would be highly attractive for soybean (MVCR = 3) and even more attractive for maize (MVCR = 5.6). This equates to just 2.2% of the current price of NPK fertilizer. Our results demonstrate that BSF frass can enhance crop yield and profitability in maize and soybean systems. While further research is necessary before these findings can be translated into large-scale application, this study establishes a foundational reference and framework toward achieving that goal. In Madagascar, intensifying maize production systems with organic fertilizers could increase yield and limit habitat destruction. Improved soybean productivity could reduce reliance on importation and diversify smallholder cropping systems. These findings highlight the promise of

provided the original author and source are credited.

**Data availability statement:** All data, codes, files to generate the manuscript can be accessed at https://github.com/ramiadantsoa/OptimBSFF. The raw data and codes are also included in this submission.

**Funding:** The author(s) received no specific funding for this work.

**Competing interests:** The authors have declared that no competing interests exist.

insect-derived fertilizers as a practical pathway toward sustainable intensification in low-yield agricultural contexts.

---

## Introduction

Agricultural production must approximately double by 2050 to meet projected global food demand [1], while simultaneously aligning with sustainable development goals. Although agricultural extensification—land clearing to increase cultivated area—remains possible in some low-income countries, this strategy is environmentally unsustainable, carries substantial ecological costs such as deforestation and habitat loss [2]. Furthermore, extensification has contributed little to recent global yield gains [3]. In contrast, agricultural intensification through improved water and nutrient use efficiency, adoption of appropriate technologies, and deployment of improved crop varieties offers a pathway to increase production while minimizing environmental pressures [3–5]. While yields in high-input systems are approaching ceilings, intensification in low-yield regions could close yield gaps between high- and low-producing countries, particularly in the Global South, and help meet growing food demand [3].

Nutrient deficiency remains a principal constraint on crop productivity in low-income countries, making increased fertilizer use a central strategy for yield intensification. Although inorganic fertilizers such as NPK are available in sufficient quantities, their adoption is often limited by poor economic returns in low-income settings, where fertilizer-to-crop price ratios are high. For example, nitrogen application rates average only 1.9 kg ha$^{-1}$ in Uganda, compared with a global mean of 139 kg ha$^{-1}$ [6]. In addition, the long-term and often excessive use of mineral fertilizers in high-input systems has been associated with soil degradation, nutrient imbalances, water pollution, and greenhouse gas emissions [7–13]. Together, these economic and environmental constraints underscore the need for affordable and locally available nitrogen sources whose application rates can be optimized under realistic economic conditions. These constraints are particularly acute in Madagascar, where fertilizer access remains limited and yield gaps are large.

The rapid expansion of insect farming presents a promising opportunity [e.g., 14–16]. Frass–a nutrient-rich organic residue produced during insect rearing–has emerged as a potential organic fertilizer. Numerous studies have evaluated the agronomic potential of frass [e.g., 17–19]. Among insect-based fertilizers, frass derived from Black Soldier Fly (*Hermetia illucens*) farming has received particular attention [20]. Key agronomic properties of BSF frass, including phytotoxicity, effects on plant growth and development, nutrient mineralization dynamics, and nitrogen use efficiency are well studied [21–23]. Practically, BSF frass can achieve crop yield comparable to that of conventional NPK fertilizers [18,24]. Such foundational agronomic assessments are essential prerequisites for scaling up insect-based fertilizers from experimental contexts to industrial and commercial applications [25].

Black Soldier Fly (BSF) production systems can generate substantial quantities of frass relative to the volume of organic waste processed [15]. Depending on rearing conditions and feedstock composition, BSF larvae typically convert 15–25% of the

initial waste mass into larval biomass, while 40–60% remains as residual frass and substrate material [26,27]. In Madagascar, BSF rearing systems have access to large and continuous feedstock streams. Agro-industrial by-products constitute a primary source. For example, STAR Madagascar produces over 90,000 tonnes of brewery spent grain annually [28], part of which is currently valorized by EXA Feed [29,30]. A second important source is food waste, estimated at around 15% for Madagascar [31]. In urban areas, these organic waste streams have already been leveraged for industrial-scale BSF production, as illustrated by initiatives such as BSF Tamatave [32,33]. In rural areas, similar organic waste could support smaller-scale, family-operated production systems. Recent evidence indicates a high level of acceptance of insect-based products among the population [34]. In addition, the establishment of national standards authorizing BSF larvae and derived products for human consumption may further incentivize the expansion of BSF farming [35]. This growth is expected to generate increasing quantities of frass, with potential applications as an organic fertilizer in agricultural systems.

Translating results from laboratory and small-scale field experiments to extensive farming systems remains challenging. Experimental trials often strive for homogeneity in soil and environmental conditions, which rarely reflects on-farm variability. Moreover, many studies are focused on comparative assessments of a small number of frass application rates with conventional fertilizers (positive controls) or unfertilized treatments (negative controls) [22,36]. While demonstrating equivalent or superior performance is informative, it does not necessarily identify the agronomically or economically optimal application rates. Although some studies have reported monotonic or hump-shaped yield responses to increasing frass doses [21,37,38], the limited number of treatments—often fewer than five—and narrow application ranges restrict the reliability of dose–response inference. A quantitative understanding of the relationship between fertilizer dose and yield is therefore essential to optimize fertilizer use and support efficient, sustainable intensification.

In this study, we evaluated the effects of BSF frass fertilizer on crop yield to determine optimal application rates for maize (*Zea mays* L.) and soybean (*Glycine max* L.) in Madagascar. These crops are strategically important in Madagascar: demand for maize is increasing, and declining yields have driven agricultural extensification with consequent habitat loss, while soybean production remains minimal resulting in heavy reliance on imports. Current average maize yields are approximately 1.8 t ha$^{-1}$, roughly one-third of the global average, and national soybean production is limited to 46 t harvested from 78 ha, resulting in a heavy reliance on imports [39]. We conducted a large-scale field experiment following agroecological practices, with the primary objective of quantifying crop yield responses to frass application rate rather than characterizing additional properties of frass. Specifically, we asked: (i) which frass application rates maximizes crop yield; (ii) how nitrogen agronomic efficiency varies across frass application rates; (iii) how the economic profitability of BSF frass varies across fertilizer application rates and retail price scenarios. By addressing these questions across a wide range of application rates under field conditions, this study explicitly identifies both agronomic and economic optima for BSF frass use in low-yield farming systems.

## Materials and methods

### Study site and soil characteristics

The study was conducted in northern Madagascar (12.3114° S, 49.2831° E) on a 800 m$^2$ experimental field located near Antsiranana, Diana Region. The region has a tropical savanna climate (Köppen Aw), with mean annual temperature of approximately 26 °C and average annual precipitation of 1,200 mm. Rainfall is strongly seasonal, with a rainy season from November to April and a dry season from May to October.

The field site belongs to an industrial brewery and was uncultivated and abandoned for over three decades. Prior to the establishment of the experiment, three composite soil samples were collected from the site at 0–20 cm depth, homogenized, and analyzed at the national agricultural laboratory (FOFIFA, Antananarivo, Madagascar). Soil analyses indicated acidic conditions (pH = 5.86), very low total nitrogen content, a low C:N ratio, intermediate levels of total organic carbon and potassium, and relatively high available phosphorus (Table 1).

**Table 1. Soil properties. OM stands for organic matter.**

| pH | N (%) | C (%) | OM (%) | C/N | P (ppm) | K (meq/100g) | SO$_4^{2-}$ |
|---|---|---|---|---|---|---|---|
| 5.86 | 0.182 | 1.204 | 2.076 | 6.7 | 24.45 | 0.203 | 16.37 |

The experiment followed agroecological management practices [5,40–43]. No tillage was performed and no machinery was used to avoid soil compaction. All crops were grown under rainfed conditions without irrigation. On 7 January, 2025, existing surface vegetation was cleared manually, and the removed biomass was redistributed uniformly across the site as mulch. Adding mulch is a common practice in Madagascar to retain soil moisture, control temperature, and suppress weeds.

### Fertilizer and crop material

The Black Soldier Fly (BSF; *Hermetia illucens*) frass used in this study was obtained from a nearby commercial BSF production facility (EXA Feed, Antsiranana, Madagascar). Larvae were reared on brewery spent grain sourced from a local industrial brewery (STAR, Madagascar). Frass was collected following larval harvest, stockpiled, and air-dried under ambient conditions for up to five days prior to field application.

Representative subsamples of the frass were analyzed for physicochemical properties at the national agricultural laboratory (FOFIFA, Antananarivo, Madagascar). The frass had a total nitrogen content of 3.46% (by mass), a C:N ratio of 12.7, and moderate concentrations of phosphorus and potassium (Table 2). This nitrogen concentration was used to calculate nitrogen application rates, nitrogen agronomic efficiency, and all subsequent analyses (see below).

Certified seed was used for both crops. Maize (*Zea mays* L.) variety used was IRAT200, purchased from AgriVet (Antsiranana, Madagascar) and is widely used by farmers in the region. Soybean (*Glycine max* L.) variety was OC11, obtained from Honey of Madagascar (Antananarivo, Madagascar), and is commonly promoted for local production.

### Experimental design and fertilizer application

For each crop, we tested twelve frass application rates to characterize yield responses across a broad fertilizer range: a control without fertilizer (0 t ha$^{-1}$), two low rates (0.4 and 0.8 t ha$^{-1}$), seven intermediate rates increasing in 2 t ha$^{-1}$ increments (3, 5, 7, 9, 11, 13, and 15 t ha$^{-1}$), and two high, intentionally excessive rates (20 and 30 t ha$^{-1}$). The range of tested rates encompass both locally documented smallholder practice (1.2 t ha$^{-1}$; [36]) and the nationally recommended dose for maize (3.6 t ha$^{-1}$; [44]), while extending to rates consistent with or exceeding those reported in published BSF frass field trials for maize [21] and soybean [37,38]. This design ensures that our dose–response inferences are grounded in agronomically relevant contexts while providing sufficient range to characterize the full shape of the response curve. The twelve application rates were selected to cover a wide range of plausible fertilizer doses while maintaining sufficient replication; we distributed the treatments into three tiers (low, intermediate, and excessive) to prioritize higher data density within the intermediate range while still exploring a wide parameter space. Each fertilizer treatment was replicated five times per crop. In total, 60 plots were established per crop. Treatments were arranged in a randomized complete block design (RCBD).

Plots were established on 8 January, 2025. For maize, each plot measured of 2 m x 2 m, and consisted of three rows of ten equally spaced plants. Rows were spaced 80 cm apart with 20 cm between plants within rows. For soybean, each plot measured 2 m x 1 m, and contained three rows of 13 equally spaced individuals. Rows were spaced 40 cm apart with 15 cm between plants. Adjacent plots were separated by 50 cm buffer strips. To minimize edge effects, each experimental area was bordered by a 1-m wide buffer zone planted with two rows of maize or soybean and fertilized at a rate of 15 t ha$^{-1}$, corresponding to the midpoint between the highest and lowest application rates.

**Table 2. Soil properties. OM stands for organic matter.**

| Humidity (%) | pH | C (%) | OM (%) | N total (%) | C/N | P (%) | K (%) | Ca (%) | Mg (%) | Na (%) | Mn (ppm) | Cu (pppm) |
|---|---|---|---|---|---|---|---|---|---|---|---|---|
| 9.4 | 6.44 | 44.0 | 88.0 | 3.46 | 12.7 | 1.12 | 0.6 | 0.01 | 0.43 | 0.04 | 25.5 | 0.5 |

Mulch was gently pushed aside in a 10 cm band along each planting row. A shallow trench (5–6 cm depth) was opened, frass was applied evenly along the row and mixed with the soil. Frass was applied on 10 January, 2025.

## Crop management

Maize and soybean were sown manually. A seed was planted at a depth of approximately 3–4 cm, with maize planted on 14 January, 2025 and soybean on 15 January, 2025. Weed removal was performed manually to avoid the use of herbicides. At the early stage, mulch prevented the development of deep weeds roots, and gentle mulch shaking mostly killed all weeds. After germination when plant height exceeded mulch height, mulch was put back closer to the root of the plants. In addition, visible weeds were removed manually every two weeks.

Pest management initially relied on preventive, low-impact methods. A neem-based solution (*Azadirachta indica*) was prepared by infusing 1.5 kg of fresh neem leaves in 5 L of water and applied preventively every 10 days at a total volume of approximately 45 L per crop. Despite these measures, pest pressure increased the first month, necessitating targeted chemical interventions to prevent severe crop loss. In maize, plots were treated with Megalégion 44 EC at a dilution rate of two tablespoons per 15 L of water on 6 February, 2025 and 8 February, 2025 to control armyworm infestations. In soybean, plots were treated with Pyrifos 480 EC at the same dilution rate on 2 February, 2025 and 4 February, 2025 in response to increasing weevil damage. Chemical applications were applied uniformly across all plots within each crop to avoid confounding fertilizer treatment effects.

## Data collection

Because maize and soybean were sown one day apart, all measurements are reported relative to sowing date as days after sowing (DAS). Although the primary objective of the study was to quantify yield responses to fertilizer application rate, additional data were collected throughout the growing season to characterize crop establishment, vegetative growth, and reproductive development.

## Germination

Germination was assessed 16 DAS for maize and 15 DAS for soybean, following the approach used in comparable BSF frass field trials [36,37]. For each plot, the proportion of emerged seedlings relative to the number of seeds sown was recorded.

## Vegetative growth and reproductive development

In maize, vegetative growth was measured 39 DAS, following standard agronomic protocols [45]. Total plant height (from the soil surface to the apex of the tallest leaf), height to the highest fully expanded leaf collar, and the number of visible leaf collars were recorded for each plant. In soybean, vegetative growth was measured 38 DAS. Vegetative growth was quantified as canopy cover [46], estimated by measuring the maximum plant length and the perpendicular width, and computing the area of an ellipse using these two orthogonal axes.

Reproductive development was evaluated 55 DAS for maize and 54 DAS for soybean. For each plant, reproductive transition was assessed by the presence of flowers in maize and pods in soybean.

## Harvest and yield determination

Maize was harvested manually 128 DAS, when all ears had fully matured and dried in the field, following the individual-plant harvest protocol [21]. For maize, each plant within a plot was harvested individually. The number of ears per plant was recorded, and for each ear, length, diameter, and fresh weight were measured. Ears were then air-dried and manually threshed to determine dry grain weight [36]. Plant-level yield was calculated as the sum of dry grain weights across all ears produced by an individual plant.

Soybean was harvested manually between 90 and 92 DAS, once all pods had dried but before they burst open [37,38]. For each plant, pods were threshed and the weight of dry grain was recorded.

Plot-level yields were converted to tonnes per hectare (t ha$^{-1}$) based on the harvested plot area and used as the primary metric for agronomic and economic analyses. In total, we collected data from 1800 maize plants and 2340 soybean plants across all treatments.

## Data analyses

All analyses were conducted at the plot level unless otherwise specified. Proportions were used as response variables for germination and reproductive development. For growth and yield, values were averaged within plots prior to analysis.

Nine days after sowing, unusually heavy rainfall caused temporary flooding in 16 maize plots and 28 soybean plots. As the randomized complete block design (RCBD) ensured no correlation between fertilizer dose and flooding status, these plots were treated as outliers and excluded from the analysis without introducing treatment bias. This approach allowed for a more accurate characterization of the nutrient dose-response relationship by removing flood-induced experimental noise.

## Germination and growth

The effect of fertilizer application rate on germination was analyzed using generalized linear mixed-effects models (GLMMs) with a binomial error distribution, consistent with the analytical approach used in comparable field studies [36]. The application rate was set as a fixed effect whereas plot replicate was included as a random effect to account for the natural variability above. Models were fitted using the *glmer* function in the lme4 package in R.

For vegetative growth variables, multiple modeling approaches were explored, including breakpoint analyses, generalized linear models (GLMs) with appropriate error structures to account for overdispersion, and locally estimated scatterplot smoothing (LOESS) when parametric models provided poor fits. Breakpoint analyses were implemented using the *breakpoints* function in the strucchange package, and GLMs were fitted using the *glm* function. Model choice was based on structural appropriateness and visual goodness of fit rather than formal model selection criteria.

## Yield per individual and per plot

To quantify the effect of fertilizer dose on yield, nonlinear asymptotic regression models were fitted to both average yield per individual plant and total yield per plot using the self-starting asymptotic function *SSasymp*. Models estimate three parameters: the asymptotic maximum yield, the rate of increase with fertilizer application rate, and the intercept corresponding to yield without fertilizer. Parameter estimation was performed using nonlinear least squares via the *nls* function in R.

To assess potential trade-offs between vegetative growth and yield, we examined relationships between individual plant yield and vegetative growth within plots. Vegetative growth was represented by total plant height for maize and canopy cover for soybean (see above). For each plot, a linear regression was fitted between individual yield and vegetative growth, and the regression coefficient was extracted. A positive coefficient indicates that yield increases with vegetative growth, whereas a negative coefficient suggests a trade-off in which increased biomass is associated with reduced yield. Linear regressions were then used to test whether these coefficients varied systematically with fertilizer application rate.

 

## Nitrogen agronomic efficiency

Nitrogen agronomic efficiency (NAE, Equation 1) was calculated following the equation stated by Baligar et al. [47]:

$$\text{NAE(dose)} = \frac{\text{yield(dose) - yield(0)}}{\text{dose} \times 0.0346} \tag{1}$$

where dose is fertilizer application rate in t ha$^{-1}$, yield(dose) is the predicted average yield from the fitted asymptotic function above, yield(0) is the predicted yield without fertilizer (model intercept), and 0.0346 represents the nitrogen content (fraction by mass) of BSF frass (Table 2). NAE therefore has units of kilograms of additional grain yield per kilogram of nitrogen applied (kg grain kg$^{-1}$ N).

## Economic profitability

Economic performance was evaluated using the marginal value-cost ratio (MVCR) [6]. MVCR represents the ratio of the additional value of crop yield generated by fertilizer application to the cost of the fertilizer applied. MVCR (Equation 2) was calculated using the equation stated by Ragasa et al. [6]:

$$\text{MVCR} = \frac{\text{price}_Y}{\text{price}_F} \times \frac{\Delta \text{yield}}{\Delta \text{dose}} \tag{2}$$

where $\text{price}_Y$ and $\text{price}_F$ is the price of one tonne of the crop and fertilizer, respectively. $\Delta$yield represents the increase in crop yield relative to the unfertilized control, and $\Delta$dose represents the quantity of fertilizer added. yield was calculated using predicted yields from the asymptotic yield models, in our case $\Delta$dose in our case is the application rate added.

Crop prices were based on FAO statistics [48] (USD 130.4 t$^{-1}$ for maize and USD 352.4 t$^{-1}$ for soybean). Profitability was evaluated for two fertilizer application rates: (i) a locally used rate of 1.2 t ha$^{-1}$ [36], hereafter referred to as the local dose, and (ii) the dose recommended by FOFIFA [44], hereafter referred to as the recommended dose. We could not find reference for soybean (considering it is barely produced in Madagascar) so we used the same doses above. Fertilizer use was considered profitable when MVCR $\geq$ 1, acceptable for risk-averse farmers when MVCR $\geq$ 2, and highly attractive when MVCR $\geq$ 3 [49–51]. A comprehensive assessment of farm-level profitability should ideally incorporate labor costs, transport expenditures, and logistical overheads, all of which are highly context-specific [e.g., 52]. Rather than offering prescriptive management recommendations, this study explores the theoretical economic boundaries of frass application, with a specific focus on fertilizer-to-crop price ratios.

## Statistical philosophy and reproducibility

Analyses were intentionally descriptive rather than hypothesis-testing in nature. The primary objective was to characterize dose–response relationships and identify structurally appropriate models rather than to test null hypotheses. Effect sizes and associated p-values are reported where relevant and are interpreted as continuous measures of evidence rather than against fixed significance thresholds [*sensu* 53].

All analyses were conducted using R version 4.4.1 [54], and the manuscript was written using Quarto version 1.6. Contour plots for the economic analyses were generated using Mathematica version 14.3 [55]. Raw data and scripts are publicly available at: https://github.com/ramiadantsoa/OptimBSFF.

## Results

### Germination rate

The effect of BSF frass on germination differed between maize and soybean (Fig 1). In maize, germination was high in the unfertilized control (93%) but declined to 85% at the highest application rate (30 t ha$^{-1}$). There is moderate evidence that increasing fertilizer rate reduced maize germination ($\beta$ = −0.0277, p=0.036; Fig 1A).

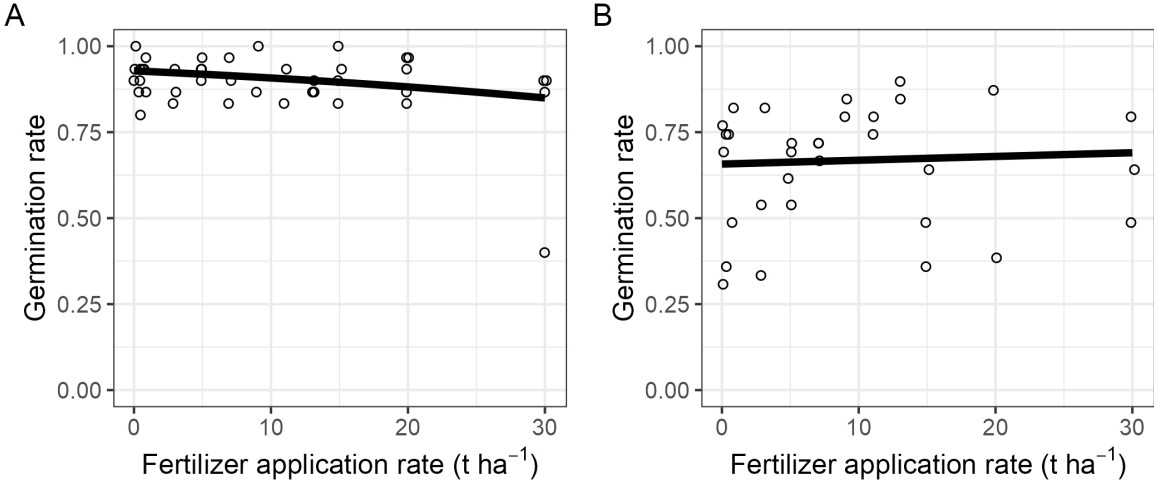

**Fig 1. Germination rate.** Proportion of seeds that germinated as function of the fertilizer rate after 16 days for maize **(A)** and 15 for soybean **(B)**. Each dot represents one plot and was jittered horizontally for visibility. The data was fitted with generalized linear mixed-effect models (black line).

In soybean, germination was lower overall (66% in unfertilized plots) and increased slightly to 69% at 30 t ha$^{-1}$. There was no evidence that fertilizer rate affected soybean germination ($\beta$ = 0, p = 0.754; Fig 1B).

The effects of fertilizer application rate on vegetative growth and reproductive development are detailed in the supporting information S1 Fig. Breakpoint analyses indicated that for maize, both plant height and the proportion of individuals reaching the reproductive stage initially increased before reaching a plateau. Conversely, breakpoint models provided a poor fit for soybean. However, LOESS smoothing suggested that while mean soybean crown cover also approached an asymptote, there was no evidence that the fertilizer application rate affected the proportion of soybean plants reaching the reproductive stage.

## Yield per individual

In maize, dry grain yield per plant increased with fertilizer rate with strong evidence of diminishing marginal returns ($\beta$ = −2.47, p < 0.001), with an estimated asymptotic yield of 182.7 g plant$^{-1}$ (Fig 2A). Across plots, early vegetative growth in maize (plant height at 39 DAS) was positively associated with final grain yield at harvest (128 DAS), as indicated by predominantly positive within-plot regression slopes. Fertilizer rate did not modify this growth–yield relationship ($\beta$ = 0.01, p = 0.556; Fig 2C).

In soybean, yield responses increased with fertilizer rate but there were no evidence of diminishing marginal returns ($\beta$ = −4.48, p = 0.439; Fig 2B). Early vegetative growth (canopy cover at 38 DAS) was positively related to final yield at harvest (92 DAS). However, unlike maize, there was moderate evidence that the growth–yield relationship declined with increasing fertilizer rate ($\beta$ = −0.0004, p = 0.012; Fig 2D), indicating that yield became less tightly coupled to early canopy development at higher application rates.

## Yield total

Total grain yield per plot increased monotonically with fertilizer rate for both crops, with clear diminishing returns at higher application rates (Fig 3AB). In maize, there was strong evidence that increasing fertilizer rate increased total yield with diminishing marginal returns ($\beta$ = −2.22, p < 0.001). The asymptotic yield was estimated at 8.6 t ha$^{-1}$, corresponding to a 1129.8% increase relative to unfertilized plots (Fig 3A). Applying the local (1.2 t ha$^{-1}$)

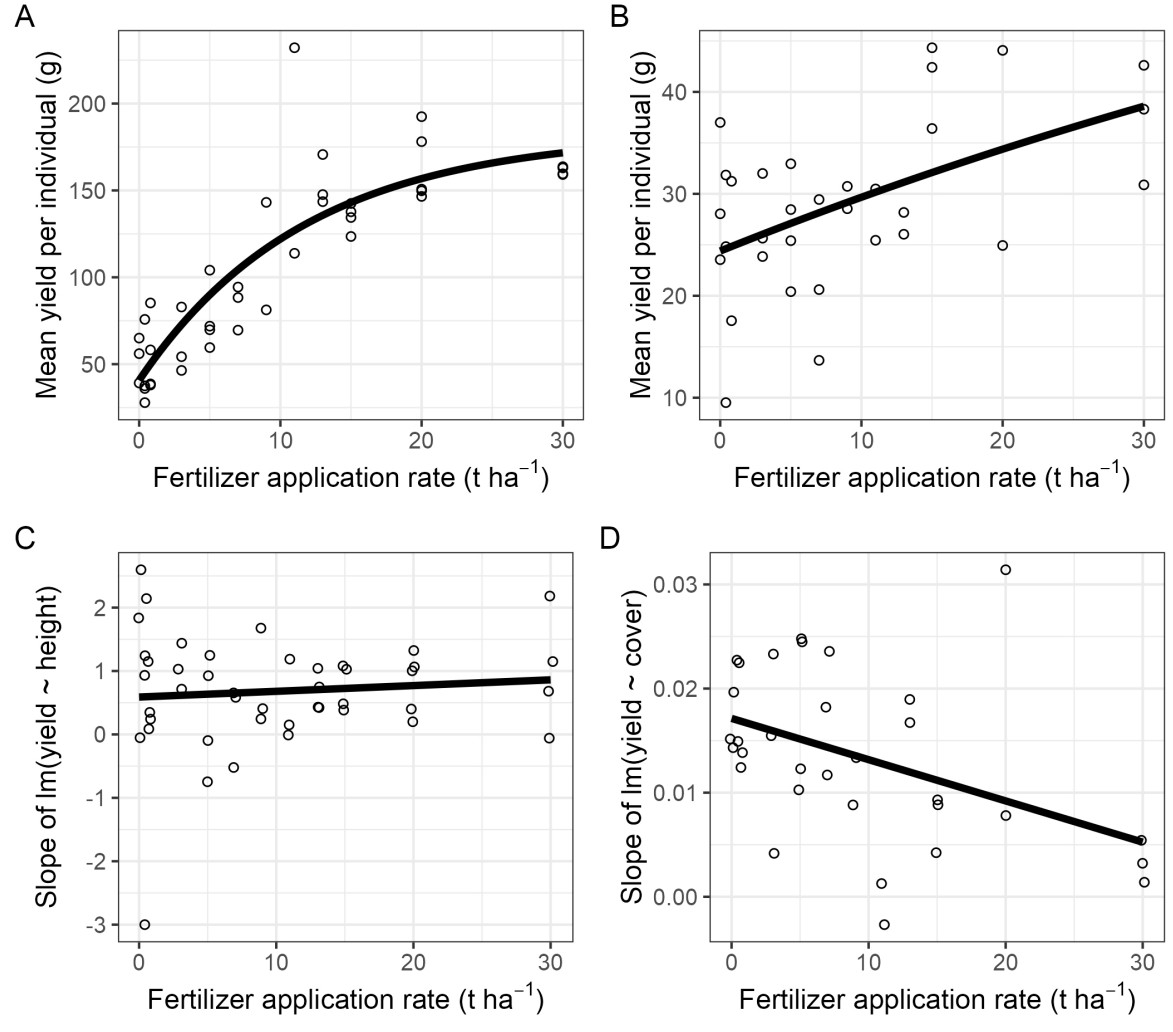

**Fig 2. Grain yield per individual.** *Top* Average dry gain yield per individual for maize **(A)** after 128 days and soybean **(B)** after 92 days as a function of fertilizer rate. Each dot represents a value from a plot. The data was fitted with asymptotic model (black line). *Bottom*: Effect of fertilizer rate on the relation between vegetative growth and yield. Each dot represents the slope of a linear regression between vegetative growth and yield for maize **(C)** and soybean **(D)**. Vegetative growth measured as height for maize and crown cover for soybean. The data was fitted with linear model (black line).

and the recommended (3.6 t ha$^{-1}$) frass doses was projected to produce maize yields of 1.67 and 3.27 t ha$^{-1}$, respectively.

In soybean, the same overall pattern was observed. Total yield increased with fertilizer rate ($\beta$ = −2.21, p = 0.002), with an estimated asymptotic yield of 4.1 t ha$^{-1}$, representing a 75% increase relative to the control (Fig 3B). Applying the local and the recommended doses was projected to produce soybean yields of 2.55 and 2.91 t ha$^{-1}$, respectively.

Nitrogen agronomic efficiency (NAE) declined with increasing fertilizer rate for both crops (Fig 3C). In maize, application of 0.4 t ha$^{-1}$ resulted in an NAE of 24.4 kg grain kg$^{-1}$ N, whereas application of 30 t ha$^{-1}$ reduced NAE to 7.33 kg grain kg$^{-1}$ N (a 70% decline).

In soybean, fertilizer addition was less efficient overall. Application of 0.4 t ha$^{-1}$ resulted in an NAE of 5.43 kg grain kg$^{-1}$ N, while 30 t ha$^{-1}$ reduced NAE to 1.62 kg grain kg$^{-1}$ N (a 70% decline; Fig 3C, dashed line).

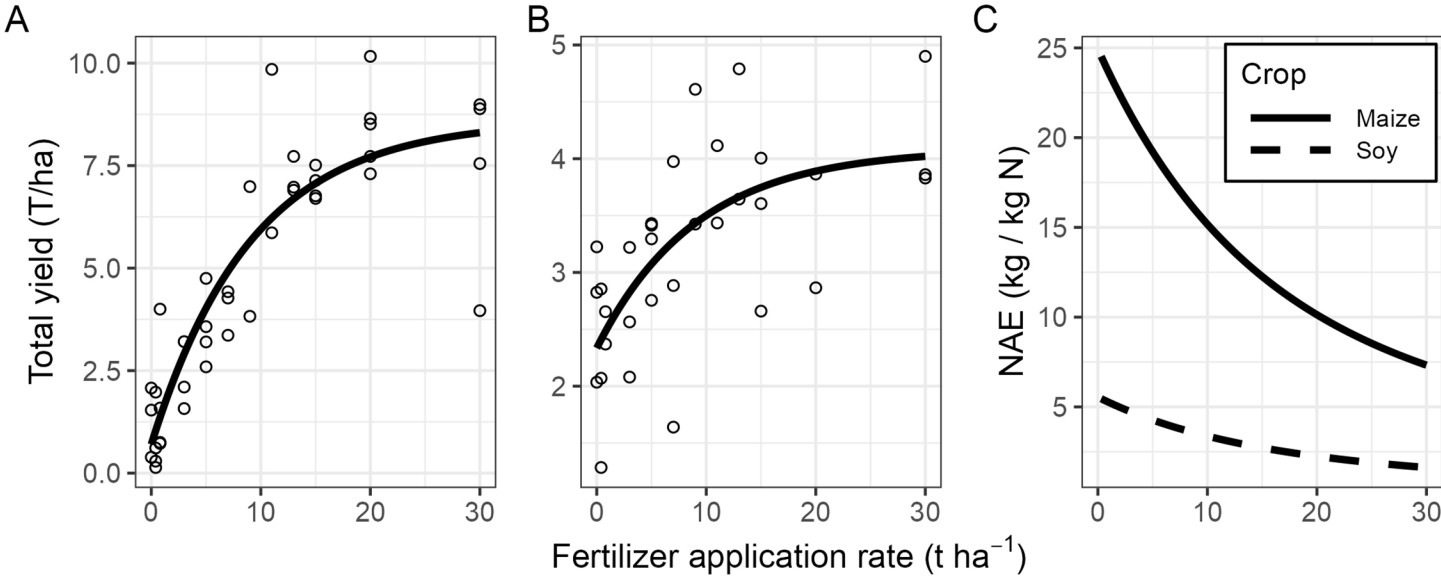

**Fig 3. Total grain yield.** Total yield per plot as a function of fertilizer rate for maize after 128 days **(A)** and soybean after 92 days **(B)**. Each dot represents a plot. The data was fitted with asymptotic model (see main text for details). **(C)** NAE: Nitrogen agronomic efficiency (kg yield / kg N) for maize (solid) and soybean (dashed).

## Economic profitability

Fig 4 shows how the marginal value–cost ratio (MVCR) varies with frass price and fertilizer rate under FAO crop prices. The profitable parameter space (non-hatched area) was smaller for soybean than for maize, indicating that soybean profitability constrained the maximum economically viable frass price when considering both crops.

Under current local fertilizer practice (1.2 t ha$^{-1}$), maize production became profitable (MVCR ≥ 1) when frass prices were below USD 52.59 t$^{-1}$ (Fig 4A). Using the recommended dose (3.6 t ha$^{-1}$), the maximum profitable frass price decreased to USD 46.2 t$^{-1}$. For soybean, frass prices needed to remain below USD 31.61 t$^{-1}$ under local practice and USD 27.75 t$^{-1}$ under the recommended rate (Fig 4B). Thus, to ensure profitability across both crops under FAO crop prices, frass prices should not exceed approximately USD 31.61 t$^{-1}$; at this soybean-constrained price, the maize is acceptable in both local and recommended doses, MVCR = 2.32 and 1.92, respectively.

Using the highly attractive threshold for farmers (MVCR ≥ 3), allowable frass prices decreased further. For maize, the maximum acceptable frass price was USD 26.3 t$^{-1}$ under local practice and USD 23.1 t$^{-1}$ under the recommended rate (Fig 4A). For soybean, frass prices needed to remain below USD 15.8 t$^{-1}$ and USD 13.87 t$^{-1}$ under local and recommended rates, respectively (Fig 4B). At a frass price of USD 15.8 t$^{-1}$, maize production became highly attractive using local and recommended doses (MVCR = 5.65 and 4.85, respectively).

In the supporting information, we provide alternative calculation based on optimal fertilization rate and net gain as a function of yield price and frass retail price S2 Fig.

## Discussion

This study quantified the relationship between Black Soldier Fly (*Hermetia illucens*) frass fertilizer application rate and dry grain yield of maize and soybean under field conditions in northern Madagascar. Grain yield increased monotonically with increasing frass application rate for both crops, even at intentionally excessive rates, but with diminishing marginal returns. Asymptotic yields corresponded to approximately 4.8 and 6.8 times current national average yields. With the

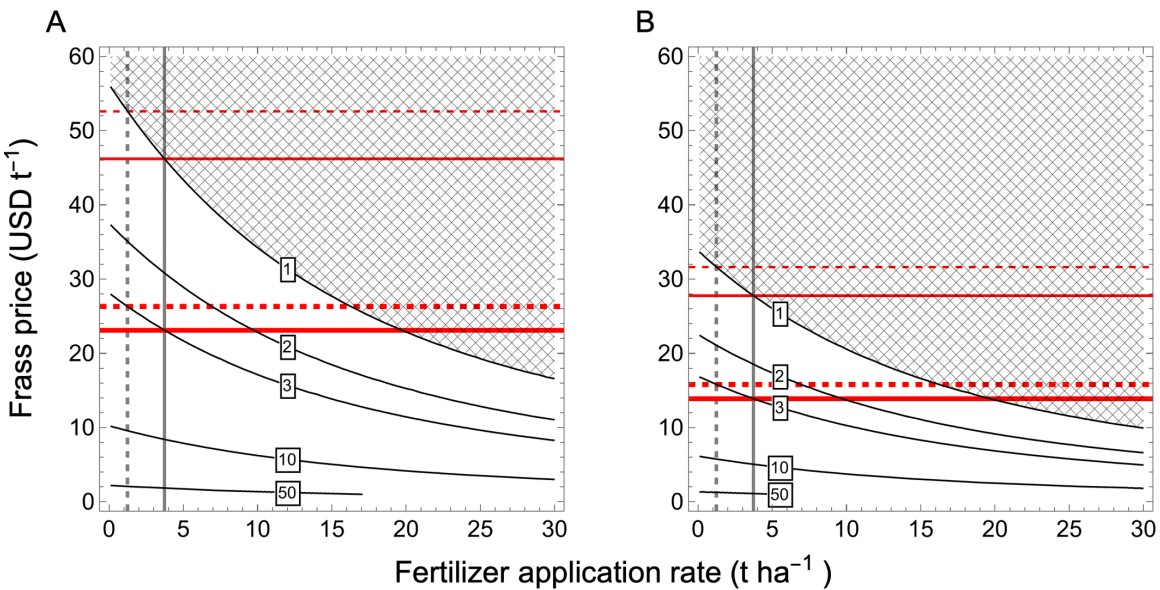

**Fig 4. Marginal value-cost ratio.** Marginal value-cost ratio as a function of fertilizer rate (x-axis) and the price of frass (y-axis) for maize **(A)** and soybean **(B)**. Each contour line represents an isocline with equal marginal value-cost ratio. The hashed area means that the combination of fertilizer rate and frass price are not profitable (MVCR < 1). The dashed and solid vertical show the local practice and FOFIFA recommended fertilizer rate for maize, respectively. The thin upper and thick lower dashed horizontal line red show the frass price that would give a marginal value-cost ratio of 1 and 3 if the local practice is applied. The upper and lower solid line is based on FOFIFA recommendation with a MVCR of 1 and 3.

inferred dose-yield relationship, we explored the trade-off between frass prices and application rates to optimize the profitability of BSF for practical large-scale use. Frass could be sold between USD 52.59 and 13.87 t$^{-1}$, where the upper value represents a profitable threshold for maize (MVCR = 1) but not for soybean, whereas the lower represents highly attractive scenario (MVCR ≥ 3) using the FOFIFA recommended dose for both soybean and maize. Consequently, the kilogram of nitrogen from BSF frass could be 8.4% to 2.2% of the retail price of mineral fertilizer (NPK 11 22 16, FDA [56]). Eventually, the optimal application rate and retail price is context dependent. This work provides a framework and reference for such optimization, and unequivocally demonstrates the agronomic and economic potential of BSF frass in Madagascar.

## Phytotoxicity

Because insect-derived fertilizers are relatively novel, assessing potential adverse effects prior to widespread adoption is essential [19]. A primary concern is phytotoxicity, which can inhibit seed germination or early seedling development. Previous studies have reported phytotoxic effects of frass [e.g., 36]. In contrast, we observed negligible phytotoxic effects of BSF frass on germination in both crops. Germination increased slightly in soybean across the fertilizer gradient, while maize germination declined by approximately 8% at the highest application rate (30 t ha$^{-1}$) relative to the unfertilized control. This reduction remains below commonly accepted thresholds for agronomic relevance [>20%, 57].

Unlike studies using fresh frass [e.g., 36], the frass used here was air-dried for up to five days prior to application, which may have reduced labile or phytotoxic compounds. While composting has been proposed as an additional stabilization step, it may be unnecessary for BSF frass and impractical at large scale. Indeed, Beesigamukama et al. [17] reported that BSF frass is relatively mature compared with other insect-derived fertilizers. Overall, phytotoxicity depends on interacting factors—including insect species, larval feed, application rate, timing, and crop species [57]—and our results indicate that BSF frass is compatible with maize and soybean production under Malagasy conditions.

## Yield response to frass application

A central objective of this study was to determine whether crop yield exhibits a hump-shaped response to frass application rate, which would permit identification of a simple yield-maximizing optimum: fertilizer rate that would maximize yield. While yield declines at high frass application rate have been reported in some systems [e.g., 58,38], we observed no evidence of yield penalties even at excessive application rates of 20–30 t ha$^{-1}$. Instead, yields increased monotonically with diminishing returns, consistent with results reported for soybean by Kisego et al. [37] and for maize by Beesigamukama et al. [21]. Importantly, our study spanned a much wider range of fertilizer rates than most previous work, strengthening the conclusion that positive yield responses to BSF frass are common and that yield suppression at high doses is not inevitable.

The observed diminishing returns indicate a saturating response, with yields approaching an asymptote at high application rates. While breakpoint or plateau analyses are sometimes used to define agronomic optima, such approaches provided poor fits in our case and risk imposing arbitrary thresholds not supported by the data. In contrast, asymptotic modeling provided a parsimonious and biologically interpretable description of the dose–response relationship. Using this approach, we estimated maximum attainable yields of 8.6 t ha$^{-1}$ for maize and 4.1 t ha$^{-1}$ for soybean. These values are comparable to average yields in high-input regions such as the European Union, exceed global averages by approximately 30–90%, and represent four- to six-fold increases over current national averages in Madagascar [6,39]. Together, these results demonstrate that BSF frass can deliver substantial yield gains without compromising early establishment or vegetative growth, and has strong potential to narrow yield gaps between low- and high-yield regions.

Because our study focused on inferring dose-response relationships, data collection prioritized yield over traditional agronomic parameters like life-history traits and physiological responses. Nevertheless, examining the relationship between yield and a snapshot of early growth reveals distinct responses between maize and soybean. In maize, yield saturated at both the individual plant and total plot levels, indicating that the crop had reached a biological limit [45]. In contrast, soybean yield saturated at the plot level but not at the individual plant level. Mathematically, this discrepancy arose because reduced plant density at higher fertilizer doses capped total yield (see S3 Fig). Biologically, it reflects a well-documented trade-off in soybean, where density-driven competition for light prompts resource allocation toward vegetative canopy growth at the expense of pod development [46,59,60]. This finding is consistent with our field observations of dense canopy overlap at high doses and the lack of relationship between fertilizer dose and reproductive development (S1 Fig). Furthermore, the relationship between growth and yield becomes weaker at high doses, confirming that spatial competition decoupled vegetative growth from reproductive output. Our narrow 15-cm inter-plant spacing—compared to the 20 cm reported in other studies [e.g., 61]—likely exacerbated this effect, though varietal differences may also play a role [62]. Ultimately, while our primary aim was optimizing fertilizer application rates, these findings highlight planting density as a critical co-factor for yield optimization [e.g., 45].

Reported NAE values vary widely across cropping systems and environments, and low fertilizer responsiveness has been documented in many smallholder contexts [63]. Solofondranohatra et al. [36] reported higher NAE values at low frass application rates, largely because yields in unfertilized controls were very low, inflating efficiency estimates. In our study, unfertilized soybean yields were relatively high–likely due to the relatively good initial condition of the soil as the land was abandoned for over three decades–resulting in lower NAE values. Our NAE estimates are consistent with global averages reported by Tilman et al. [1], reinforcing their plausibility. For soybean, lower NAE is consistent with the crop's biological nitrogen fixation capacity and aligns with previous recommendations [38].

We acknowledge that certain aspects of this experimental design warrant caution regarding the generalizability of the inferred dose-response relationships. Specifically, our experimental site exhibited high baseline soil phosphorus (P) levels. In a comparable study utilizing BSF frass for maize in southeastern Madagascar, poor initial soil fertility led to complete crop failure in unfertilized control plots; however, their yield at an equivalent frass dose was slightly higher than ours (1.98 vs. 1.6 t ha$^{-1}$) [36]. Despite differences in farming practices—most notably, the use of irrigation in the latter study—these

results indicate that BSF frass supplies sufficient essential nutrients to support crop growth in a variety of soils. Furthermore, the consistent yield increases observed in our high-dose treatments suggest an absence of significant nutrient antagonism (such as P-induced zinc or iron deficiency), which frequently limits yields in high-P soils [64,65]. Consequently, we hypothesize that applying this dose-response model to sites with normal or lower baseline P levels would primarily lower the baseline yield (the intercept) without fundamentally altering the general shape of the response curve or lowering the asymptotic maximum yield [66].

Additionally, our study was conducted over a single growing season, limiting inferences regarding longer-term agronomic effects, such as residual nitrogen availability or progressive changes in soil nutrient dynamics [67]. Multi-year trials are necessary to strengthen these conclusions, particularly concerning the cumulative soil-health benefits of organic amendments like BSF frass [22]. Given the agroecological, pedological, climatic heterogeneity of Madagascar [68], predicted yields are likely to vary geographically, especially at lower fertilizer application rates where baseline soil properties exert a stronger influence. Nonetheless, our empirical approach is arguably more transferable to real-world smallholder conditions than highly parameterized, strictly controlled agronomic models. Ultimately, this study provides an extensive and crucial baseline on the efficacy of BSF frass for maize and soybean production in Madagascar.

## Economic profitability and implication for adoption

Fertilizer cost remains a primary barrier to agricultural intensification in low-income countries. In Madagascar, the nitrogen-equivalent cost of mineral NPK fertilizer often exceeds the economic returns achievable at prevailing crop prices. The kg of N from chemical fertilizer is currently at USD 18.3 kg$^{-1}$ N [56], and low responsiveness could further prevent adoption [6]. Using marginal value–cost ratio analysis, we showed that BSF frass remains economically viable across a wide range of price scenarios. Under conservative assumptions based on FAO crop prices, frass prices between USD 52.59 and 13.87 t$^{-1}$ were profitable to highly attractive, depending on crop and application rate. This corresponds to nitrogen prices of approximately USD 0.4–1.5 kg$^{-1}$ N, a fraction of that of mineral NPK fertilizer in Madagascar, and mineral fertilizers reported elsewhere in sub-Saharan Africa [6].

Our economic calculations represent a simplified model of reality. For instance, FAO commodity prices often diverge from local market fluctuations. However, the mathematical structure of the Marginal Value Cost Ratio (MVCR) implies that if the crop price increases by a given factor, the frass price can also increase by the same factor without changing the final MVCR value. Therefore, the frass price can simply be adjusted proportionally to the crop price to maintain consistent economic viability for the farmer. Our findings also indicate that the profitability threshold of soybean effectively dictates the maximum retail price of BSF frass. Although soybean commands a higher market price USD 352.4 compared to maize 130.4, it exhibits a weaker agronomic response to fertilization, characterized by lower nitrogen agronomic efficiency (NAE) and a modest 1.5-fold yield increase over the unfertilized control. This unsurprising response is consistent since soybean is nitrogen-fixing legume. Consequently, if regional agricultural priorities favor maize production, the frass pricing model could be optimized based on maize profitability rather than soybean.

Furthermore, a comprehensive economic assessment must incorporate expenditures, such as labor, transport, and additional agricultural inputs [52]. These costs are highly context-dependent [6,23,68]. For smallholder farmers, family labor is frequently unaccounted for as a direct cash expense. Land tenure systems, such as sharecropping, further complicate economic models by fundamentally altering how input costs (e.g., tools, labor) and final yields are divided between tenants and landowners. Conversely, large-scale industrial farming operates under entirely different financial structures and economies of scale. Parallel complexities exist on the supply side, as BSF frass can be sourced from either decentralized smallholder systems or large-scale industrial facilities. Despite these inherent limitations, the primary strength of this study lies in establishing empirical dose-response curves (caveats regarding their broad generalizability above) and providing a foundational framework to evaluate the mutual economic viability of frass application for both producers and farmers in Madagascar.

## Conclusion

This study adds robust field-based evidence supporting the use of BSF frass as an effective fertilizer for sustainable intensification in low-input agricultural systems. By moving beyond small-scale agronomic trials and explicitly quantifying yield responses and economic profitability, we provide practical guidance for large-scale adoption. In regions where fertilizer accessibility limits productivity, BSF frass offers a rare combination of agronomic effectiveness, economic affordability, and environmental sustainability. When integrated with complementary practices such as improved crop varieties and conservation agriculture [3,5], BSF frass has the potential to play a transformative role in closing yield gaps, reducing pressure on natural ecosystems, and advancing food security and rural livelihoods in Madagascar and similar contexts.

## Supporting information

**S1 Fig. Vegetative growth and development stage.** Vegetative growth after 39 days (top) and proportion of individual with flower (bottom) as a function of quantity of frass amended after 55 days for maize (A), and after 54 days for soybean (B). Top panels: each point represents the average height of maize (A) and crown cover of soy (B) across 5 replicates. Bottom panels: each dot represents one plot and was jittered horizontally for visibility. The data was fitted with breakpoint analyses (black line), alternative model where chosen when the fit was poor (dashed black line) (see main text for details). (TIF)

**S2 Fig. Optimal fertilizer application rate and net gain.** Optimal frass fertilizer application rate (top) and net gain (bottom) for maize (left) and soybean (right) as a function of the price of frass (x-axis) and the price of yield (y-axis). The vertical dashed lines represent how much one tonne of frass would cost if based on the N equivalent from NPK. The dashed horizontal lines show the price according to FAO data. The red and magenta solid lines represent local and FOFIFA recommended dose, respectively. The hashed area show regions where adding fertilizer is not profitable. (TIF)

**S3 Fig. Number of individuals with yield for maize (A), and for soybean (B).** Each dot represents one plot. The data was fitted with LOESS (black line). (TIF)

## Acknowledgments

We thank the agronomy and biodiversity students from the University of Antsiranana for their essential assistance with fieldwork and data collection. We are grateful to the staff of EXA Feed (Antsiranana) for logistical support and access to Black Soldier Fly frass, and to FOFIFA (Antananarivo) for soil and fertilizer analyses. We are grateful to Mulki Salendra Kusumah and two anonymous reviewers for their valuable feedback that helped improve this manuscript.

## Author contributions

**Conceptualization:** Tanjona Ramiadantsoa, Marc Philippart, Vincent Lucas.

**Formal analysis:** Tanjona Ramiadantsoa.

**Funding acquisition:** Vincent Lucas, Brian L. Fisher.

**Methodology:** Tanjona Ramiadantsoa, Anthonely Amady, Marc Philippart, Marcellio Tombozara.

**Resources:** Anthonely Amady, Marcellio Tombozara, Vincent Lucas.

**Writing – original draft:** Tanjona Ramiadantsoa.

**Writing – review & editing:** Tanjona Ramiadantsoa, Anthonely Amady, Brian L. Fisher.

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
