## [Decision Letter · Decision Letter 0]

2 Mar 2026

PONE-D-26-06219Optimizing Black Soldier Fly frass fertilizer rates for maize and soybean production in MadagascarPLOS One

Dear Dr. Ramiadantsoa,

Thank you for submitting your manuscript to PLOS ONE. After careful consideration, we feel that it has merit but does not fully meet PLOS ONE’s publication criteria as it currently stands. Therefore, we invite you to submit a revised version of the manuscript that addresses the points raised during the review process.

If applicable, we recommend that you deposit your laboratory protocols in protocols.io to enhance the reproducibility of your results. Protocols.io assigns your protocol its own identifier (DOI) so that it can be cited independently in the future. For instructions see: https://journals.plos.org/plosone/s/submission-guidelines#loc-laboratory-protocols. Additionally, PLOS ONE offers an option for publishing peer-reviewed Lab Protocol articles, which describe protocols hosted on protocols.io. Read more information on sharing protocols at . Additionally, PLOS ONE offers an option for publishing peer-reviewed Lab Protocol articles, which describe protocols hosted on protocols.io. Read more information on sharing protocols at https://plos.org/protocols?utm_medium=editorial-email&utm_source=authorletters&utm_campaign=protocols..

We look forward to receiving your revised manuscript.

Kind regards,

Dave Mangindaan

Academic Editor

PLOS One

**Journal Requirements:**

https://journals.plos.org/plosone/s/file?id=wjVg/PLOSOne_formatting_sample_main_body.pdf andand

3. We note that you have referenced (All analyses above used FAO crop prices, which are approximately half of prevailing local market prices (unpublished data) and (However, in other experiments conducted by us, increasing frass application rate did not affect germination (unpublished data), indicating that any transient effects are likely context dependent.) which has currently not yet been accepted for publication. Please remove this from your References and amend this to state in the body of your manuscript: (ie “Bewick et al. [Unpublished]”) as detailed online in our guide for authors

**Additional Editor Comments:**

Please kindly revise according to the assessment and suggestions of the reviewers. Please also download the comments in the attachment of the review via logging into EditorialManager.com system.  In particular, I also agree to the suggestion to add recent studies related to the effect of various feed in the growing BSF larvae, especially when some of the references are older than 2020s, such as:

•    Green Technologies and Sustainability 2 (2024) 100112 https://doi.org/10.1016/j.grets.2024.100112

•    Scientific African 27 (2025) e02626 https://doi.org/10.1016/j.sciaf.2025.e02626

Thank you.

Assoc. Prof. Dave Mangindaan, PhD, MRSC, AMIChemE, ASEAN Eng.

Reviewers' comments:

Reviewer's Responses to Questions

**Comments to the Author**

1. Is the manuscript technically sound, and do the data support the conclusions?

Reviewer #1: Yes

Reviewer #2: Yes

Reviewer #3: Partly

2. Has the statistical analysis been performed appropriately and rigorously? 

Reviewer #1: Yes

Reviewer #2: Yes

Reviewer #3: Yes

3. Have the authors made all data underlying the findings in their manuscript fully available?

Reviewer #1: Yes

Reviewer #2: Yes

Reviewer #3: Yes

4. Is the manuscript presented in an intelligible fashion and written in standard English?

Reviewer #1: Yes

Reviewer #2: Yes

Reviewer #3: Yes

5. Review Comments to the Author

Reviewer #1: Comments for Author,

Thank you for the submission. Even the paper is interesting, I have some comments before accepted this paper .

Specific comments

Extend introduction with more refs..

Give more information regarding data analyses.

Discussion is superficial. Please give more data.

Best Regards

Reviewer #2: This is a timely and relevant study that addresses a critical gap in the practical application of insect-based fertilizers. The large-scale field trial and the focus on both agronomic and economic optima are significant strengths. However, several points require clarification, correction, or further development to enhance the manuscript's rigor and impact before publication.

1. You have introduced two significant uncontrolled variables: flooding and dolomite application. By not including "dolomite" as a fixed factor in your statistical models and by not stratifying or blocking for the "flooding" effect, you cannot separate the true effect of the frass rate from the effects of liming and waterlogging stress. The statement that this provides "conservative estimates" (Line 170) is not accurate. It creates noise and potentially bias, making it harder to detect true treatment effects and potentially skewing the dose-response curve. This is not a matter of simply mentioning it in the supplementary material. The primary analysis must account for these factors. I would strongly suggest to:

a. For dolomite: Include it as a fixed effect in all your mixed-effects models

b. For flooding: You should either (a) exclude flooded plots from the primary analysis if they are few and the flooding was clearly an outlier event, or (b) include a binary "flooded" variable as a fixed effect in the models. Presenting only the pooled analysis leaves the results open to criticism that the observed patterns are driven by site heterogeneity, not the treatment itself.

2. NAE is a descriptive metric for a specific rate. Multiplying it by the predicted yield to find a maximum is mathematically unusual and biologically ambiguous. The "NAE-adjusted yield" is not a standard agronomic or physiological parameter. What does this product represent? It appears to heavily penalize higher yields, inevitably driving the "optimum" to a lower rate. The authors state this balances "yield gains against efficient nutrient use" (Line 395). However, the most economically relevant optimization already exists in your Marginal Value-Cost Ratio (MVCR) analysis, which directly uses crop and fertilizer prices. NAE is an efficiency metric, not a direct proxy for environmental impact (like nitrogen leaching potential) or economic return. Using it to create a new "optimum" is redundant and conceptually less clear than the MVCR. I strongly advise removing the "NAE-adjusted yield" and the "NAE-optimal dose" (Lines 229-230, 344-345, 395-408, and Figure 3D). The agronomic optimum is best described by the plateau of the yield response curve (the asymptote). The economic optimum is best described by the MVCR analysis. The NAE analysis is useful for describing how efficiency declines (Figure 3C), which is a valuable finding in itself. Focus the narrative on these two clear, complementary concepts: biophysical potential and economic feasibility.

3. The economic analysis needs to be reframed. Acknowledge these limitations explicitly (you do in Lines 449-452, but it's too late). The current analysis shows the maximum allowable frass price at the farm gate. It does not demonstrate that using these optimal rates is profitable in reality. The conclusion that frass prices "between USD 15 and 97 t⁻¹ were acceptable to profitable" (Line 414) is therefore a theoretical upper bound, not a practical recommendation. The discussion must be tempered to reflect that for the farmer, the relevant cost is "frass price + transport + labor."

4. Expand on the physiological trade-off observed in soybean (growth vs. yield) and the importance of the soil's initial P status for interpreting the results.

5. Soil Context (Table 1): The soil has "relatively high available phosphorus" (P = 24.45 ppm). This is an important point. The positive yield response you see is largely a response to nitrogen (and perhaps potassium/micronutrients) from the frass, not phosphorus. This should be highlighted, as it means the results are most applicable to soils with adequate P. On P-deficient soils, the response to frass might be even greater or qualitatively different.

6. Lines 194-199 (Growth-Yield Relationship): The interpretation of the slopes in Figure 2C and 2D is interesting. For maize (2C), the flat line suggests that the positive link between early vigor and final yield is maintained at all frass rates. For soybean (2D), the declining slope is a key finding. It suggests that at high frass rates, high early vegetative cover does not necessarily translate into high grain yield, possibly due to luxury vegetative growth at the expense of pod filling. This point is mentioned but could be elaborated on in the discussion as a potential physiological trade-off.

This study has the potential to be a valuable contribution to the field of sustainable agriculture. Addressing these points, particularly the experimental design confounding and the economic analysis, will significantly strengthen its conclusions and practical relevance.

Reviewer #3: Major Comment

This manuscript presents a well-designed field study evaluating the agronomic and economic performance of BSF frass as an organic fertilizer for maize and soybean in Madagascar. The study addresses a critical constraint to agricultural intensification in low-income systems, namely fertilizer affordability, and integrates asymptotic yield modeling with NAE and MVCR analyses. The results demonstrate consistent yield increases without evidence of yield penalties at high application rates and provide an agronomic–economic framework that could inform fertilizr recommendations under resource-constrained conditions. Beyond its immediate findings, the study has practical relevance for circular bioeconomy strategies, offering evidnce that locally produced organic fertilizers such as BSF frass may help reduce dependency on imported mineral fertilizers, enhance nutrient recycling, and contribute to narrowing yield gaps in smallholder systems.

However, the experimental design indicates that dolomite was systematically applied to half of the plots within each frass rate, effectively creating a factorial structure, yet dolomite was not included as a fixed factor in the main statistical model, nor were frass × dolomite interactions formally explained. Given the well-established effects of liming on soil pH, nutrient availability, and nitrogen dynamics, excluding this factor may influence interpretation of the dose–response relationships. In addition, flooding events were reported but not clearly incorporated or analytically justified in the primry model. For consistency between experimental design and statistical inference, these factors should either be explicitly modeled (including dolomite and its interaction with frass rate in a mixed-effects framework) or clearly reported as tested and found non-significant, with appropriate model specifications and statistical evidence. That said, I remain neutral at this stage, as relevant data are presented in the Supplementary Material. I would therefore be interested to see whether the second reviewer shares similar concerns before recommending a more definitive course of action.

In addition, I consider it important to highlight that several sections, particularly within the Methods, contain limited citation of supporting literature. Methodological decisions should ideally be grounded in established experimental design principles or prior validated approaches, yet the manuscript provides relatively few references to justify key analytical and experimental choices. This limited citation base may compromise methodological traceability and future reproducibility. I therefore strongly encourage the authors to substantially strengthen the reference support, especially in the Methods section, to enhance transparency, scientific rigor, and reproducibility.

Based on the considerations outlined above, I regret that at this stage I must recommend major revision. This decision reflects concerns regarding analytical consistency and methodological transparency rather than the overall scientific merit of the work. I strongly encourage the authors to address the issues raised, as the findings are important and have significant potential impact; with appropriate revisions, this study could make a valuable contribution to the literature and to broader discussions on sustainable fertilizr strategies in agricultural systems.

I have prepared a table to provide a more detailed overview of the specific points that require improvement based on my feedback, which is presented below as Table 1.

Tabel 1. Minor Comment

No. Line(s) Comments

1 26-36 Please add information on frass productivity derived from similar waste streams or from the local region, to illustrate product abundance and its potential as a fertilizer alternative.

2 107-109 Could you clarify whether the selection of these frass application rates is based on established references or prior studies? Please provide the supporting sources.

3 117-127 Please provide references to support the technical procedures used for maize and soybean planting.

4 154, 157, 167, 194, 207 Please provide references to support the technical procedures used for this study.

5 224-225, 235-236 Add definitions/explanations for the equation parameters, especially 'ton', 'ton h-1', etc. These should be clearly described.

6 224 Add a citation to “(Equation 1)” in the text after “Nitrogen agronomic efficiency (NAE) was calculated as”.

7 223 The formulation of nitrogen agronomic efficiency (NAE) requires clarification. Please provide the literature reference supporting the equation used, present the formula explicitly in its complete mathematical form (including the denominator), and clearly state all units for each term to avoid ambiguity. In addition, the definition of “NAE-adjusted yield” should be written as an explicit equation, with its corresponding units, and supported by an appropriate methodological or theoretical justification. Clear and transparent presentation of these formulas is essential for reproducibility and reader comprehension.

8 234 Add a citation to “(Equation 1)” in the text after “Economic performance was evaluated using the marginal value-cost ratio (MVCR), calculated as”.

9 234-239 Please provide an appropriate reference to support this formula.

10 295, 300 The reported application rate of 0.04 t ha⁻¹ associated with the highest NAE appears inconsistent with the described treatment range (0.4 t ha⁻¹ to 30 t ha⁻¹) and seems unusually low, therefore please clarify whether this value is correct, model-derived, or a typographical error.

11 325-326 Please provide a verifiable source for the local crop prices described as “approximately double the FAO values,” as these data are currently cited as unpublished; given that profitability thresholds scale directly with crop price, transparent documentation of these local price assumptions is essential for evaluating the robustness of the economic conclusions, and these data should be included either directly in the manuscript or as supplementary material.

12 397-399 Please provide the supporting literature for this statement, as no reference is currently cited to substantiate the statement.

13 427-428 The statement “substantially lower than those for cattle manure, mineral NPK fertilizer in Madagascar, and mineral fertilizers reported elsewhere in sub-Saharan Africa” lacks an explicit citation for cattle manure prices specifically in Madagascar. Reference [6] appears to cover mineral fertilizer cost comparisons but does not substantiate the claim regarding cattle manure. Please provide an appropriate literature or data source supporting the cattle manure price comparison, and include this reference directly in the manuscript or, if necessary, in the supplementary material.

14 Fig. 4 In Figure 4, the contour labels (1, 2, 3, 10, 50) overlap with key areas of the plot and partially obscure the underlying data, making interpretation difficult; please adjust the placement of these labels to improve clarity and readability.

15 Fig. 4C Figure 4C is not mentioned in the manuscript text. Please ensure it is appropriately referred to and its role in the study is clearly described

16 References Please add more references from the recent five years to show the novelty/new progress brought by this study. There are still some from 2002, 2005, 2008, 2011, 2012, 2019.

Sincerely,

Mulki Salendra Kusumah

6. PLOS authors have the option to publish the peer review history of their article (what does this mean?). If published, this will include your full peer review and any attached files.). If published, this will include your full peer review and any attached files.

.

Reviewer #1: No

Reviewer #2: No

Reviewer #3: **Yes:** Mulki Salendra KusumahMulki Salendra Kusumah

---

## [Author Response · Author response to Decision Letter 1]

22 Mar 2026

Response to Reviewers

Manuscript title: Optimizing Black Soldier Fly frass fertilizer rates for maize and soybean production in Madagascar

Dear Editor,

Thank you for the thoughtful comments and for the opportunity to revise our manuscript. We have substantially revised the manuscript in response to the reviewers’ suggestions.

The most important change concerns the analytical treatment of dolomite application and flooding. After re-evaluating the dataset, we concluded that plots receiving dolomite were confounded by unintended differences in implementation, including a difference in the interval between fertilization and sowing. Because we could not confidently attribute any observed differences specifically to dolomite, we removed all dolomite-treated plots from the revised analyses. We also removed plots affected by the anomalous flooding event. Although this reduced the dataset, the main conclusions remained unchanged. We believe this substantially simplifies the manuscript and strengthens the inference regarding the effect of frass application rate.

We also agree with Reviewer #2 that the “NAE-adjusted optimum” was conceptually awkward. We therefore removed that analysis and all associated interpretation, and now focus on two clearer dimensions of inference: biophysical response and economic profitability.

In addition, we extensively revised the Discussion, strengthened the Methods, and added 27 references, including 11 published within the last five years. We also added a paragraph in the Introduction on frass productivity from organic waste streams, clarified the rationale for the fertilizer dose design, revised the economic interpretation to emphasize that the analysis represents a theoretical upper bound on economically viable frass prices, and improved methodological transparency throughout.

We hope that these revisions have improved the manuscript and made it suitable for publication in PLOS ONE.

Sincerely,

Tanjona Ramiadantsoa

Reviewer #1

Thank you for the submission. Even the paper is interesting, I have some comments before accepted this paper. Specific comments: Extend introduction with more refs. Give more information regarding data analyses. Discussion is superficial. Please give more data.

Response: We thank the reviewer for these helpful comments. In response, we expanded the Introduction with additional references, clarified the Data Analyses section, and substantially revised the Discussion to provide more interpretation and context for the results. These revisions were also informed by the more specific suggestions of Reviewers #2 and #3.

Reviewer #2

This is a timely and relevant study that addresses a critical gap in the practical application of insect-based fertilizers. The large-scale field trial and the focus on both agronomic and economic optima are significant strengths. However, several points require clarification, correction, or further development to enhance the manuscript's rigor and impact before publication.

Response: We thank the reviewer for the positive assessment of the study and for the constructive suggestions. We address each point below.

1. Dolomite application and flooding were not adequately accounted for in the original analyses.

You have introduced two significant uncontrolled variables: flooding and dolomite application. By not including dolomite as a fixed factor in your statistical models and by not stratifying or blocking for the flooding effect, you cannot separate the true effect of the frass rate from the effects of liming and waterlogging stress. The primary analysis must account for these factors.

Response: We agree that these factors required clearer treatment. After re-evaluating the dataset, we concluded that the dolomite treatment could not be interpreted cleanly because it was confounded by unintended implementation differences, most notably a three-day difference in the interval between fertilizer application and sowing. We therefore removed all dolomite-treated plots from the revised analyses rather than presenting an analysis that might be misleading. We also removed plots affected by the anomalous flooding event. In the revised dataset, 44 maize plots and 32 soybean plots were retained. We re-ran all analyses, and the main conclusions remained unchanged, except that soybean individual yield no longer clearly approached an asymptote. Overall, germination and yield responses were slightly stronger in the revised analyses. We believe this decision simplifies the manuscript and strengthens the inference regarding the effect of frass application rate.

2. NAE-adjusted yield should be removed.

NAE is a descriptive metric for a specific rate. Multiplying it by the predicted yield to find a maximum is mathematically unusual and biologically ambiguous. The NAE-adjusted optimum is conceptually less clear than the MVCR analysis.

Response: We agree and have removed the NAE-adjusted yield analysis and all associated interpretation from the revised manuscript. We now use NAE only as a descriptive efficiency metric and focus the main narrative on two clearer and complementary concepts: biophysical response and economic profitability.

3. The economic analysis should be reframed as a theoretical upper bound rather than a practical farm-level recommendation.

The current analysis shows the maximum allowable frass price at the farm gate. It does not demonstrate that using these rates is profitable in reality because labor, transport, and other costs are not included.

Response: We agree and revised the Methods, Results, and Discussion to make clear that the analysis provides a theoretical upper bound on economically viable frass prices rather than a complete farm-level profitability assessment. In the Methods, we now state: “A comprehensive assessment of farm-level profitability should ideally incorporate labor costs, transport expenditures, and logistical overheads, all of which are highly context-specific (e.g., Mamadou et al., 2020). Rather than offering prescriptive management recommendations, this study explores the theoretical economic boundaries of frass application, with a specific focus on fertilizer-to-crop price ratios.” We also added two paragraphs in the Discussion clarifying that FAO prices are only a reference point, that MVCR rescales proportionally with crop price, and that labor, transport, land tenure, and production structure all influence realized profitability.

4. Expand on the physiological trade-off observed in soybean and the role of initial soil phosphorus.

Expand on the physiological trade-off observed in soybean (growth vs. yield) and the importance of the soil's initial P status for interpreting the results.

Response: We expanded the Discussion with two new paragraphs addressing these points. First, we discuss the divergence between individual-level and plot-level soybean yield responses and interpret this pattern as a density-related trade-off in which vegetative growth becomes decoupled from reproductive output at high fertilizer rates. Second, we discuss the implications of the relatively high baseline soil phosphorus at the study site and clarify that the observed responses likely reflect primarily nitrogen and possibly other nutrients supplied by the frass. We also discuss how responses might differ in soils with lower baseline phosphorus.

5. Soil context and phosphorus availability.

The positive yield response is likely driven largely by nitrogen rather than phosphorus because the soil already had relatively high available phosphorus. This should be highlighted.

Response: We agree and now address this explicitly in the Discussion as part of the expanded paragraph on soil phosphorus context and generalizability.

6. Elaborate on the soybean growth–yield relationship shown in Figure 2D.

The declining slope in soybean suggests that high early vegetative cover does not necessarily translate into high grain yield at high frass rates, possibly due to a trade-off favoring vegetative growth over pod filling.

Response: We agree and expanded the Discussion accordingly. We now interpret the weakening growth–yield relationship at high soybean fertilizer rates as evidence that spatial competition and canopy expansion decouple vegetative growth from reproductive output under dense planting conditions.

Reviewer #3

This manuscript presents a well-designed field study evaluating the agronomic and economic performance of BSF frass as an organic fertilizer for maize and soybean in Madagascar. The study addresses a critical constraint to agricultural intensification in low-income systems, namely fertilizer affordability, and integrates asymptotic yield modeling with NAE and MVCR analyses. The results demonstrate consistent yield increases without evidence of yield penalties at high application rates and provide an agronomic–economic framework that could inform fertilizer recommendations under resource-constrained conditions.

Response: We thank the reviewer for this positive assessment and for the detailed and constructive comments. We address the major and minor points below.

Major comment. Dolomite and flooding should be handled more explicitly.

The experimental design indicates that dolomite was systematically applied to half of the plots, yet dolomite was not included as a fixed factor in the main statistical model. Flooding events were also not clearly incorporated or analytically justified in the primary model.

Response: We thank the reviewer for this important point. We shared this concern, especially after the same issue was raised by Reviewer #2. After re-evaluating the dataset, we removed all dolomite-treated plots and all flooded plots from the revised analyses. The revised manuscript now reflects this reduced dataset throughout. The main results remained unchanged. We believe this change makes the inference cleaner and avoids over-interpreting a confounded dolomite treatment.

Methods citations and transparency.

Several sections, particularly within the Methods, contain limited citation of supporting literature. The manuscript would benefit from stronger reference support for methodological decisions.

Response: We thank the reviewer for this suggestion. We added references in the Methods where appropriate, especially for agroecological management, agronomic efficiency, and economic evaluation. We also clarified procedural details directly in the text when no single citation was especially appropriate. We hope these revisions improve methodological transparency and reproducibility.

1. Add information on frass productivity derived from similar waste streams or from the local region.

Response: We added a new paragraph in the Introduction describing the expected quantity of frass generated by BSF systems processing organic waste streams, with specific reference to agro-industrial substrates such as brewery spent grain. This addition is intended to better illustrate the potential abundance of frass as a locally available fertilizer alternative.

2. Clarify whether the selected frass application rates were based on established references or prior studies.

Response: We clarified the rationale for the selected application rates in the Methods. Briefly, the twelve doses were chosen to sample a broad range of the parameter space while concentrating more observations in the intermediate range, where we expected the most informative part of the dose–response curve. The design was also constrained by available field area and replication requirements.

3–4. Provide references to support the technical procedures used for planting and for other procedures in the study.

Response: We added references where appropriate and clarified the procedures in the Methods. Some measurements, such as germination, vegetative growth, and reproductive status, are standard agronomic observations and were included primarily to complement the main yield-focused analyses.

5. Clarify equation parameters and units, including t ha⁻¹.

Response: We clarified the notation and units associated with all equation terms in the revised manuscript, including the definition of t ha⁻¹.

6. Add a citation to the NAE equation.

Response: We added a citation for the NAE equation (Baligar et al., 2001).

7. Clarify the NAE formulation and define all variables and units.

Response: We clarified the NAE equation, added the supporting citation, and explicitly defined all variables and units in the text. We also removed the NAE-adjusted yield analysis in response to Reviewer #2, which resolved the reviewer’s additional concern regarding that derived metric.

8–9. Add a citation to the MVCR equation and support the formula.

Response: We added a citation for the MVCR equation (Ragasa et al., 2025) and retained supporting references for commonly used profitability thresholds (Chamberlin et al., 2021; Sheahan et al., 2013; Theriault et al., 2018).

10. Clarify the reported application rate associated with the highest NAE.

Response: We agree that this value could be confusing. It was model-derived rather than one of the experimental treatment levels. To avoid ambiguity, we replaced it with 0.4 t ha⁻¹, which corresponds to the lowest applied treatment rate. This change does not alter the interpretation.

11. Provide a verifiable source for local crop prices described as approximately double the FAO values.

Response: We agree that the local market price statement was insufficiently documented. We therefore removed that unsupported comparison from the revised manuscript and retained the FAO-based analysis, which is fully referenced. We also added discussion clarifying how the MVCR framework can be rescaled if local crop prices differ from FAO values.

12. Add supporting literature for the statement on the negative effects of excessive fertilizer use.

Response: We added supporting citations for the environmental consequences of excessive fertilizer use, including Carpenter et al. (1998), Guo et al. (2010), and Vitousek et al. (1997).

13. Provide a citation for the cattle manure price comparison in Madagascar.

Response: We removed the comparison with cattle manure in Madagascar because we could not provide a sufficiently verifiable citation. We retained the comparison with mineral NPK fertilizer and now cite a government-backed source for the current price.

14. Adjust Figure 4 contour labels to improve readability.

Response: Done. The contour labels in Figure 4 were repositioned to improve clarity and reduce overlap.

15. Figure 4C is not mentioned in the manuscript text.

Response: We removed the cattle manure results, and Figure 4 now presents only the FAO-based analyses discussed in the main text.

16. Add more references from the last five years.

Response: We added 27 new references, including 11 published within the last five years. We aimed to balance recent literature with older seminal papers that remain important for framing the study.

Journal requirements and editor comments

1. Please ensure that the manuscript meets PLOS ONE style requirements, including file naming.

Response: We reviewed the manuscript files and formatting against the PLOS ONE guidelines and will submit the revision using the required file structure and file names.

2. Please provide additional information regarding permits obtained for the work.

Response: We revised the Methods to clarify field site access and permitting information, including the approving authority where applicable.

3. Please remove references to unpublished work from the reference list and instead cite them in the text as unpublished.

Response: We revised the manuscript accordingly and removed unpublished items from the reference list where required by the journal guidelines.

4. Please include captions for Supporting Information files at the end of the manuscript and update in-text citations.

Response: We added captions for the Supporting Information files at the end of the manuscript and updated the corresponding in-text citations.

5. Please evaluate reviewer-suggested citations for relevance.

Response: We reviewed the suggested publications and incorporated relevant recent references where appropria

---

## [Decision Letter · Decision Letter 1]

8 Apr 2026

PONE-D-26-06219R1Optimizing Black Soldier Fly frass fertilizer rates for maize and soybean production in MadagascarPLOS One

Dear Dr. Ramiadantsoa,

Thank you for submitting your manuscript to PLOS ONE. After careful consideration, we feel that it has merit but does not fully meet PLOS ONE’s publication criteria as it currently stands. Therefore, we invite you to submit a revised version of the manuscript that addresses the points raised during the review process.

Please make sure to obtain Reviewer #3's comments/annotations in the attachment, via EditorialManager platform. The detailed comments provided by Reviewer #3 are very important to the improvement of this submission.  In addition, Reviewer #3 also highlighted that fresher references are needed. As a researcher of BSF myself, some of them (from 2022 and 2024) might be: https://doi.org/10.3390/agriculture14020205https://doi.org/10.3390/su142113993 The articles are of open access publication, and therefore the authors can freely read and decide to cite them or not.  Regards,Assoc. Prof. Dave Mangindaan, PhD, MRSC, AMIChemE, ASEAN Eng. 

If applicable, we recommend that you deposit your laboratory protocols in protocols.io to enhance the reproducibility of your results. Protocols.io assigns your protocol its own identifier (DOI) so that it can be cited independently in the future. For instructions see: https://journals.plos.org/plosone/s/submission-guidelines#loc-laboratory-protocols. Additionally, PLOS ONE offers an option for publishing peer-reviewed Lab Protocol articles, which describe protocols hosted on protocols.io. Read more information on sharing protocols at . Additionally, PLOS ONE offers an option for publishing peer-reviewed Lab Protocol articles, which describe protocols hosted on protocols.io. Read more information on sharing protocols at https://plos.org/protocols?utm_medium=editorial-email&utm_source=authorletters&utm_campaign=protocols..

As the corresponding author, your ORCID iD is verified in the submission system and will appear in the published article. PLOS supports the use of ORCID, and we encourage all coauthors to register for an ORCID iD and use it as well. Please encourage your coauthors to verify their ORCID iD within the submission system before final acceptance, as unverified ORCID iDs will not appear in the published article. *Only* the individual author can complete the verification step; PLOS staff the individual author can complete the verification step; PLOS staff *cannot* verify ORCID iDs on behalf of authors.verify ORCID iDs on behalf of authors.

We look forward to receiving your revised manuscript.

Kind regards,

Dave Mangindaan

Academic Editor

PLOS One

Journal Requirements:

Reviewers' comments:

Reviewer's Responses to Questions

**Comments to the Author**

1. If the authors have adequately addressed your comments raised in a previous round of review and you feel that this manuscript is now acceptable for publication, you may indicate that here to bypass the “Comments to the Author” section, enter your conflict of interest statement in the “Confidential to Editor” section, and submit your "Accept" recommendation.

Reviewer #1: All comments have been addressed

Reviewer #2: All comments have been addressed

Reviewer #3: All comments have been addressed

2. Is the manuscript technically sound, and do the data support the conclusions?

Reviewer #1: Yes

Reviewer #2: Yes

Reviewer #3: Partly

3. Has the statistical analysis been performed appropriately and rigorously? 

Reviewer #1: Yes

Reviewer #2: Yes

Reviewer #3: Yes

4. Have the authors made all data underlying the findings in their manuscript fully available?

Reviewer #1: Yes

Reviewer #2: Yes

Reviewer #3: Yes

5. Is the manuscript presented in an intelligible fashion and written in standard English?

Reviewer #1: Yes

Reviewer #2: Yes

Reviewer #3: Yes

6. Review Comments to the Author

Reviewer #1: Paper could be accepted for the publication. So, I can suggest for the publication.

Best Regards.

Reviewer #2: The authors have performed a thorough revision. The most critical changes: removing the confounded dolomite and flood-affected plots and eliminating the conceptually weak "NAE-adjusted yield" analysis, demonstrate a strong commitment to scientific rigor and clarity. The manuscript is now more focused, with a cleaner narrative centered on biophysical response and economic profitability.

The authors have not only made the requested changes but have gone further to simplify and strengthen the manuscript. The revised version is significantly improved in terms of methodological transparency, conceptual clarity, and interpretive caution.

I would, however, suggest for the authors to conduct a thorough peer review to check the terminologies and check if there are any typographical errors to improve the manuscript flow and readability. After that, I believe the manuscript is fit for publication. Congratulations.

Reviewer #3: Thank you for the revisions and for carefully addressing the previous comments. I appreciate the improvements made, even though some aspects required removal, and I acknowledge that the manuscript has improved overall. However, several points still require further clarification and refinement to strengthen the manuscript. Please consider the following detailed comments (highlighted in yellow for clarity) to further improve the quality, transparency, and robustness of the study.

1. Add information on frass productivity derived from similar waste streams or from the local region.

Response: We added a new paragraph in the Introduction describing the expected quantity of frass generated by BSF systems processing organic waste streams, with specific reference to agro-industrial substrates such as brewery spent grain. This addition is intended to better illustrate the potential abundance of frass as a locally available fertilizer alternative.

Comment: Thank you for the clarification and the additional paragraph. However, the references provided primarily justify frass yield potential based on BSF bioconversion efficiency. To strengthen the argument regarding real-world availability, I encourage the authors to include supporting data from official and region-specific sources (e.g., national statistics agencies or government reports) that quantify the production of relevant organic waste streams (such as agro-industrial residues) in the study area. This would provide a more robust and context-specific basis for assessing the practical abundance and scalability of frass as a fertilizer alternative.

2. Clarify whether the selected frass application rates were based on established references or prior studies.

Response: We clarified the rationale for the selected application rates in the Methods. Briefly, the twelve doses were chosen to sample a broad range of the parameter space while concentrating more observations in the intermediate range, where we expected the most informative part of the dose–response curve. The design was also constrained by available field area and replication requirements.

Comment: Thank you for clarifying the rationale behind the selection of application rates. I understand the constraints and the logic of your experimental design. However, I would kindly request that you also provide references to previous studies or established fertilization practices for this crop. This will help ensure that the tested rates are relatable to common field applications and farmer practices, thereby strengthening the practical relevance of your findings.

3–4. Provide references to support the technical procedures used for planting and for other procedures in the study.

Response: We added references where appropriate and clarified the procedures in the Methods. Some measurements, such as germination, vegetative growth, and reproductive status, are standard agronomic observations and were included primarily to complement the main yield-focused analyses.

Comment: Thank you for adding references and clarifying the procedures. However, I noticed that the citation was only provided at the beginning of the Methods section (“The experiment followed agroecological management practices [5, 26–29]”). For greater clarity and traceability, it would be more effective to include specific references within each relevant subsection, for example, under Germination, Vegetative growth and reproductive development, Harvest and yield determination, and Germination and growth. This will allow readers to easily follow and verify the procedures against prior studies, thereby strengthening the methodological rigor of your manuscript.

5. Clarify equation parameters and units, including t ha⁻¹.

Response: We clarified the notation and units associated with all equation terms in the revised manuscript, including the definition of t ha⁻¹.

Comment: Thank you for the clarification.

6. Add a citation to the NAE equation.

Response: We added a citation for the NAE equation (Baligar et al., 2001).

Comment: Thank you for adding the citation for the NAE equation. The reference is appropriate; however, the phrase “(Equation 1)” should still be explicitly included in the text. The intended format is: “Nitrogen agronomic efficiency (NAE) was calculated as (Equation 1):” followed by the citation (e.g., Baligar et al., 2001). Please see, for example, the formatting used in [https://doi.org/10.1371/journal.pone.0332046]. Including both the equation label and the reference will improve clarity and consistency for readers.

7. Clarify the NAE formulation and define all variables and units.

Response: We clarified the NAE equation, added the supporting citation, and explicitly defined all variables and units in the text. We also removed the NAE-adjusted yield analysis in response to Reviewer #2, which resolved the reviewer’s additional concern regarding that derived metric.

Comment: Thank you for the clarification. This revision is reasonable and can be accepted.

8–9. Add a citation to the MVCR equation and support the formula.

Response: We added a citation for the MVCR equation (Ragasa et al., 2025) and retained supporting references for commonly used profitability thresholds (Chamberlin et al., 2021; Sheahan et al., 2013; Theriault et al., 2018).

Comment: Thank you for adding the citation for the MVCR equation and supporting references. However, as indicated in my earlier point (see Comment 6), the phrase “(Equation XX)” should still be explicitly included in the text after “Economic performance was evaluated using the marginal value-cost ratio (MVCR), calculated as”. This formatting, together with the citation, will ensure clarity and consistency for readers.

10. Clarify the reported application rate associated with the highest NAE.

Response: We agree that this value could be confusing. It was model-derived rather than one of the experimental treatment levels. To avoid ambiguity, we replaced it with 0.4 t ha⁻¹, which corresponds to the lowest applied treatment rate. This change does not alter the interpretation.

Comment: Thank you for the clarification. This revision is reasonable and can be accepted.

11. Provide a verifiable source for local crop prices described as approximately double the FAO values.

Response: We agree that the local market price statement was insufficiently documented. We therefore removed that unsupported comparison from the revised manuscript and retained the FAO-based analysis, which is fully referenced. We also added discussion clarifying how the MVCR framework can be rescaled if local crop prices differ from FAO values.

Comment: Thank you for the clarification. This revision is reasonable and can be accepted.

12. Add supporting literature for the statement on the negative effects of excessive fertilizer use.

Response: We added supporting citations for the environmental consequences of excessive fertilizer use, including Carpenter et al. (1998), Guo et al. (2010), and Vitousek et al. (1997).

Comment: Thank you for adding the supporting citations. The references provided are relevant, however, it would further strengthen the manuscript if more recent studies could also be included alongside the classic works.

13. Provide a citation for the cattle manure price comparison in Madagascar.

Response: We removed the comparison with cattle manure in Madagascar because we could not provide a sufficiently verifiable citation. We retained the comparison with mineral NPK fertilizer and now cite a government-backed source for the current price.

Comment: Thank you for the clarification. This revision is reasonable and can be accepted.

14. Adjust Figure 4 contour labels to improve readability.

Response: Done. The contour labels in Figure 4 were repositioned to improve clarity and reduce overlap.

Comment: Thank you for revising Figure 4. The repositioning of the contour labels has clearly improved readability, and the graph is now much easier to interpret.

15. Figure 4C is not mentioned in the manuscript text.

Response: We removed the cattle manure results, and Figure 4 now presents only the FAO-based analyses discussed in the main text.

Comment: Thank you for the clarification.

16. Add more references from the last five years.

Response: We added 27 new references, including 11 published within the last five years. We aimed to balance recent literature with older seminal papers that remain important for framing the study.

Comment: Thank you for adding the additional references, including several recent ones. However, the manuscript still appears to rely heavily on citations older than five years. I leave it to the editor’s discretion to determine whether the current balance between recent and older references is acceptable or if further adjustment is required.

7. PLOS authors have the option to publish the peer review history of their article (what does this mean?). If published, this will include your full peer review and any attached files.). If published, this will include your full peer review and any attached files.

.

Reviewer #1: No

Reviewer #2: No

Reviewer #3: No

---

## [Author Response · Author response to Decision Letter 2]

15 Apr 2026

6. Review Comments to the Author

Reviewer #1: Paper could be accepted for the publication. So, I can suggest for the publication.

Response: We appreciate the reviewer’s positive evaluation.

Best Regards.

Reviewer #2: The authors have performed a thorough revision. The most critical changes: removing the confounded dolomite and flood-affected plots and eliminating the conceptually weak "NAE-adjusted yield" analysis, demonstrate a strong commitment to scientific rigor and clarity. The manuscript is now more focused, with a cleaner narrative centered on biophysical response and economic profitability.

The authors have not only made the requested changes but have gone further to simplify and strengthen the manuscript. The revised version is significantly improved in terms of methodological transparency, conceptual clarity, and interpretive caution.

I would, however, suggest for the authors to conduct a thorough peer review to check the terminologies and check if there are any typographical errors to improve the manuscript flow and readability. After that, I believe the manuscript is fit for publication. Congratulations.

Response: We thank the reviewer for their valuable comments, which significantly improved the manuscript. We have expanded and updated the citations (see Reviewer #3) and confirmed the consistency and accuracy of the terminology.

Reviewer #3: Thank you for the revisions and for carefully addressing the previous comments. I appreciate the improvements made, even though some aspects required removal, and I acknowledge that the manuscript has improved overall. However, several points still require further clarification and refinement to strengthen the manuscript. Please consider the following detailed comments (highlighted in yellow for clarity) to further improve the quality, transparency, and robustness of the study.

1. Add information on frass productivity derived from similar waste streams or from the local region.

Response: We added a new paragraph in the Introduction describing the expected quantity of frass generated by BSF systems processing organic waste streams, with specific reference to agro-industrial substrates such as brewery spent grain. This addition is intended to better illustrate the potential abundance of frass as a locally available fertilizer alternative.

Comment: Thank you for the clarification and the additional paragraph. However, the references provided primarily justify frass yield potential based on BSF bioconversion efficiency. To strengthen the argument regarding real-world availability, I encourage the authors to include supporting data from official and region-specific sources (e.g., national statistics agencies or government reports) that quantify the production of relevant organic waste streams (such as agro-industrial residues) in the study area. This would provide a more robust and context-specific basis for assessing the practical abundance and scalability of frass as a fertilizer alternative.

Response: We have rewritten the entire paragraph to better explain the potential of BSF production systems in Madagascar. We have included two new data points on the quantities of industrial and domestic waste that could be used as feedstock. As BSF farming is a relatively recent practice in Madagascar, there are currently no published data on the actual number of BSF farms.

The paragraph now reads:

Black Soldier Fly (BSF) production systems can generate substantial quantities of

frass relative to the volume of organic waste processed [15]. Depending on rearing

conditions and feedstock composition, BSF larvae typically convert 15–25% of the initial

waste mass into larval biomass, while 40–60% remains as residual frass and substrate

material [26, 27]. In Madagascar, BSF rearing systems have access to large and

continuous feedstock streams. Agro-industrial by-products constitute a primary source.

For example, STAR Madagascar produces over 90,000 tons of brewery spent grain

annually [28], part of which is currently valorized by EXA Feed [29, 30]. A second

important source is food waste, estimated at around 15% for Madagascar [31]. In urban

areas, these organic waste streams have already been leveraged for industrial-scale BSF

production, as illustrated by initiatives such as BSF Tamatave [32, 33]. In rural areas,

similar organic waste could support smaller-scale, family-operated production systems.

Recent evidence indicates a high level of acceptance of insect-based products among the

population [34]. In addition, the establishment of national standards authorizing BSF

larvae and derived products for human consumption may further incentivize the

expansion of BSF farming [35]. This growth is expected to generate increasing quantities

of frass, with potential applications as an organic fertilizer in agricultural systems.

2. Clarify whether the selected frass application rates were based on established references or prior studies.

Response: We clarified the rationale for the selected application rates in the Methods. Briefly, the twelve doses were chosen to sample a broad range of the parameter space while concentrating more observations in the intermediate range, where we expected the most informative part of the dose–response curve. The design was also constrained by available field area and replication requirements.

Comment: Thank you for clarifying the rationale behind the selection of application rates. I understand the constraints and the logic of your experimental design. However, I would kindly request that you also provide references to previous studies or established fertilization practices for this crop. This will help ensure that the tested rates are relatable to common field applications and farmer practices, thereby strengthening the practical relevance of your findings.

Response: We agree that contextualizing the tested rates against established references strengthens the Methods. We have added a sentence to the "Experimental design and fertilizer application" subsection explicitly linking our rate range to prior studies and local practice. The low end of our design (0.4–1.2 t ha⁻¹) captures rates comparable to documented smallholder practice in Madagascar ([22], Solofondranohatra et al. 2025) and the minimum doses reported in field-scale BSF frass trials for maize ([15], Beesigamukama et al. 2020). The intermediate range spans the nationally recommended dose for maize (3.6 t ha⁻¹; FOFIFA, [32]) and includes rates consistent with prior trials for soybean ([23], Kigeso et al. 2025; [24], Yudistira et al. 2024). The upper rates (20–30 t ha⁻¹) intentionally exceed published recommendations and prior trial ranges to explore the upper limits of the agronomic response curve.

Text changes in the first paragraph of "Experimental design and fertilizer application":

“....The range of tested rates encompass both locally documented smallholder practice (1.2 t

ha-1; [36]) and the nationally recommended dose for maize (3.6 t ha-1; [44]), while

extending to rates consistent with or exceeding those reported in published BSF frass

field trials for maize [21] and soybean [37, 38]. This design ensures that our

dose–response inferences are grounded in agronomically relevant contexts while

providing sufficient range to characterize the full shape of the response curve….”

3–4. Provide references to support the technical procedures used for planting and for other procedures in the study.

Response: We added references where appropriate and clarified the procedures in the Methods. Some measurements, such as germination, vegetative growth, and reproductive status, are standard agronomic observations and were included primarily to complement the main yield-focused analyses.

Comment: Thank you for adding references and clarifying the procedures. However, I noticed that the citation was only provided at the beginning of the Methods section (“The experiment followed agroecological management practices [5, 26–29]”). For greater clarity and traceability, it would be more effective to include specific references within each relevant subsection, for example, under Germination, Vegetative growth and reproductive development, Harvest and yield determination, and Germination and growth. This will allow readers to easily follow and verify the procedures against prior studies, thereby strengthening the methodological rigor of your manuscript.

Response: We have added these references in each of subsection below and revised the text:

Germination Subsection

Add citations to Solofondranohatra et al. ([36]) and Kigeso et al. ([37]), which use the same germination assessment approach (proportion of emerged seedlings at ~15–16 DAS) in BSF frass trials on the same crops.

"Germination was assessed 16 DAS for maize and 15 DAS for soybean, following the

approach used in comparable BSF frass field trials [36, 37]. For each plot, the proportion

of emerged seedlings relative to the number of seeds sown was recorded”.

Vegetative Growth and Reproductive Development Subsection

Add citation to Zhang et al. ([45]) for maize plant height measurement protocols, and Patjaiko et al. ([46]) for soybean canopy assessment.

Maize: "In maize, vegetative growth was measured 39 DAS, following standard agronomic protocols [45]. Total plant height..."

Soybean: "In soybean, vegetative growth was measured 38 DAS. Vegetative growth was quantified as canopy cover [46], estimated by measuring the maximum plant length and the perpendicular width ..."

Harvest and Yield Determination Subsection

Add citations to Beesigamukama et al. ([21]) for the maize harvest protocol (individual plant harvest, ear measurements, air-drying, threshing) and Kigeso et al. ([37]) or Yudistira et al. ([38]) for soybean pod threshing and dry grain weight.

Maize: "Maize was harvested manually 128 DAS, when all ears had fully matured and dried in the field, following the individual-plant harvest protocol [21]. For each plant..."

Soybean: "Soybean was harvested manually between 90 and 92 DAS, once all pods had dried but before they burst open [37, 38]."

Germination and Growth (Data Analyses) Subsection

Add a citation to Solofondranohatra et al. ([36]) as a comparable field study employing the same GLMM approach for crop germination data.

"The effect of fertilizer application rate on germination was analyzed using generalized linear mixed-effects models (GLMMs) with a binomial error distribution, consistent with the analytical approach used in comparable field studies [36]. The application rate was set as a fixed effect..."

5. Clarify equation parameters and units, including t ha⁻¹.

Response: We clarified the notation and units associated with all equation terms in the revised manuscript, including the definition of t ha⁻¹.

Comment: Thank you for the clarification.

Response: We thank the reviewer for their acknowledgment and are glad that the clarification was helpful.

6. Add a citation to the NAE equation.

Response: We added a citation for the NAE equation (Baligar et al., 2001).

Comment: Thank you for adding the citation for the NAE equation. The reference is appropriate; however, the phrase “(Equation 1)” should still be explicitly included in the text. The intended format is: “Nitrogen agronomic efficiency (NAE) was calculated as (Equation 1):” followed by the citation (e.g., Baligar et al., 2001). Please see, for example, the formatting used in [https://doi.org/10.1371/journal.pone.0332046]. Including both the equation label and the reference will improve clarity and consistency for readers.

Comment: Thank you for adding the citation for the NAE equation. The reference is appropriate; however, the phrase “(Equation 1)” should still be explicitly included in the text. The intended format is: “Nitrogen agronomic efficiency (NAE) was calculated as (Equation 1):” followed by the citation (e.g., Baligar et al., 2001). Please see, for example, the formatting used in [https://doi.org/10.1371/journal.pone.0332046]. Including both the equation label and the reference will improve clarity and consistency for readers.

Response: The text has been revised to comply with the suggested formatting:

“Nitrogen agronomic efficiency (NAE, Equation 1) was calculated following the equation stated by Baligar et al. [2001]: …”

7. Clarify the NAE formulation and define all variables and units.

Response: We clarified the NAE equation, added the supporting citation, and explicitly defined all variables and units in the text. We also removed the NAE-adjusted yield analysis in response to Reviewer #2, which resolved the reviewer’s additional concern regarding that derived metric.

Comment: Thank you for the clarification. This revision is reasonable and can be accepted.

Response: We thank the reviewer for their positive assessment and are pleased that the revision is acceptable.

8–9. Add a citation to the MVCR equation and support the formula.

Response: We added a citation for the MVCR equation (Ragasa et al., 2025) and retained supporting references for commonly used profitability thresholds (Chamberlin et al., 2021; Sheahan et al., 2013; Theriault et al., 2018).

Comment: Thank you for adding the citation for the MVCR equation and supporting references. However, as indicated in my earlier point (see Comment 6), the phrase “(Equation XX)” should still be explicitly included in the text after “Economic performance was evaluated using the marginal value-cost ratio (MVCR), calculated as”. This formatting, together with the citation, will ensure clarity and consistency for readers.

Response: The text has been revised to comply with the suggested formatting:

“MVCR (Equation 2) was calculated using the equation stated by Ragasa et al. [2025]:”

10. Clarify the reported application rate associated with the highest NAE.

Response: We agree that this value could be confusing. It was model-derived rather than one of the experimental treatment levels. To avoid ambiguity, we replaced it with 0.4 t ha⁻¹, which corresponds to the lowest applied treatment rate. This change does not alter the interpretation.

Comment: Thank you for the clarification. This revision is reasonable and can be accepted.

Response: Thank you

11. Provide a verifiable source for local crop prices described as approximately double the FAO values.

Response: We agree that the local market price statement was insufficiently documented. We therefore removed that unsupported comparison from the revised manuscript and retained the FAO-based analysis, which is fully referenced. We also added discussion clarifying how the MVCR framework can be rescaled if local crop prices differ from FAO values.

Comment: Thank you for the clarification. This revision is reasonable and can be accepted.

Response: Thank you

12. Add supporting literature for the statement on the negative effects of excessive fertilizer use.

Response: We added supporting citations for the environmental consequences of excessive fertilizer use, including Carpenter et al. (1998), Guo et al. (2010), and Vitousek et al. (1997).

Comment: Thank you for adding the supporting citations. The references provided are relevant, however, it would further strengthen the manuscript if more recent studies could also be included alongside the classic works.

Response: We have incorporated four recent references (published within the last five years) to better document the negative impacts of mineral fertilizers, including their contribution to greenhouse gas emissions. The text now reads:

“In addition, the long-term and often excessive use of mineral fertilizers in high-input systems has been associated with soil degradation, nutrient imbalances, water pollution, and greenhouse gas emissions [7–13].”

13. Provide a citation for the cattle manure price comparison in Madagascar.

Response: We removed the comparison with cattle manure in Madagascar because we could not provide a sufficiently verifiable citation. We retained the comparison with mineral NPK fertilizer and now cite a government-backed source for the current price.

Comment: Thank you for the clarification. This revision is reasonable and can be accepted.

Response: Thank you

14. Adjust Figure 4 contour labels to improve readability.

Response: Done. The contour labels in Figure 4 were repositioned to improve clarity and reduce overlap.

Comment: Thank you for revising Figure 4. The repositioning of the contour labels has clearly improved re

---

## [Decision Letter · Decision Letter 2]

18 Apr 2026

Optimizing Black Soldier Fly frass fertilizer rates for maize and soybean production in Madagascar

PONE-D-26-06219R2

Dear Dr. Ramiadantsoa,

We’re pleased to inform you that your manuscript has been judged scientifically suitable for publication and will be formally accepted for publication once it meets all outstanding technical requirements.

An invoice will be generated when your article is formally accepted. Please note, if your institution has a publishing partnership with PLOS and your article meets the relevant criteria, all or part of your publication costs will be covered. Please make sure your user information is up-to-date by logging into Editorial Manager at Editorial Manager® and clicking the ‘Update My Information' link at the top of the page. For questions related to billing, please contact  and clicking the ‘Update My Information' link at the top of the page. For questions related to billing, please contact billing support..

Kind regards,

Dave Mangindaan

Academic Editor

PLOS One

Additional Editor Comments (optional):

As Reviewers 1, 2, and 3 are satisfied with the revisions made by the authors, the manuscript can be accepted for publication. Thank you for your effort and scientific contributions.

Reviewers' comments:

Reviewer's Responses to Questions

**Comments to the Author**

1. If the authors have adequately addressed your comments raised in a previous round of review and you feel that this manuscript is now acceptable for publication, you may indicate that here to bypass the “Comments to the Author” section, enter your conflict of interest statement in the “Confidential to Editor” section, and submit your "Accept" recommendation.

Reviewer #2: All comments have been addressed

Reviewer #3: All comments have been addressed

2. Is the manuscript technically sound, and do the data support the conclusions?

Reviewer #2: Yes

Reviewer #3: Yes

3. Has the statistical analysis been performed appropriately and rigorously? 

Reviewer #2: Yes

Reviewer #3: Yes

4. Have the authors made all data underlying the findings in their manuscript fully available?

Reviewer #2: Yes

Reviewer #3: Yes

5. Is the manuscript presented in an intelligible fashion and written in standard English?

Reviewer #2: Yes

Reviewer #3: Yes

6. Review Comments to the Author

Reviewer #2: The current state of the manuscript has met my expectations. It is now fit for publication in PLOS One, congratulations.

Reviewer #3: Thank you for the thorough revisions and for carefully addressing the previous comments. I appreciate the improvements made, the manuscript is now significantly clearer and more coherent in its overall flow. In my view, the manuscript has reached a standard suitable for acceptance and can proceed to the next stage of the publication process. I would like to express my appreciation for the authors’ efforts and congratulations on this work. I hope it will provide valuable contributions to the broader community.

7. PLOS authors have the option to publish the peer review history of their article (what does this mean?). If published, this will include your full peer review and any attached files.). If published, this will include your full peer review and any attached files.

.

Reviewer #2: No

Reviewer #3: **Yes:** Mulki Salendra KusumahMulki Salendra Kusumah

---

## [Editor Report · Acceptance letter]

PONE-D-26-06219R2

PLOS One

Dear Dr. Ramiadantsoa,

I'm pleased to inform you that your manuscript has been deemed suitable for publication in PLOS One. Congratulations! Your manuscript is now being handed over to our production team.

Kind regards,

on behalf of

Assoc. Prof. Dave Mangindaan

Academic Editor

PLOS One